# Topics in Cosmology—Clearly Explained by Means of Simple Examples

Jaume de Haro [1,*] and Emilio Elizalde [2,3]

1 Departament de Matemàtiques, Universitat Politècnica de Catalunya, Diagonal 647, 08028 Barcelona, Spain
2 Institute of Space Sciences, Campus UAB, 08193 Barcelona, Spain; elizalde@ice.csic.es
3 International Laboratory for Theoretical Cosmology, TUSUR University, 634050 Tomsk, Russia
* Correspondence: jaime.haro@upc.edu

**Abstract:** This is a very comprehensible review of some key issues in modern cosmology. Simple mathematical examples and analogies are used, whenever available. The starting point is the well-known Big Bang cosmology (BBC). We deal with the mathematical singularities appearing in this theory and discuss some ways to remove them. Next, and before introducing the inflationary paradigm by means of clear examples, we review the horizon and flatness problems of the old BBC model. We then consider the current cosmic acceleration and, as a procedure to deal with both periods of cosmic acceleration in a unified way, we study quintessential inflation. Finally, the reheating stage of the universe via gravitational particle production, which took place after inflation ended, is discussed in clear mathematical terms, by involving the so-called $\alpha$-attractors in the context of quintessential inflation.

**Keywords:** cosmology; inflation; cosmic acceleration; reheating

## 1. Introduction

A long time ago, humans raised their eyes to the sky and started to try to understand everything that was around: the whole universe. Its origin and evolution are among the greatest mysteries in the history of humanity. This quest was the beginning of a new science named cosmology (from "cosmos," the Greek word for universe). One of its main purposes is to study the whole universe chronology, which is related to very primordial questions for human beings, such as "where do we come from?" and "where are we going?".

The first models to describe our universe were proposed a long time ago. Among them is the Cosmic Egg allegory that appeared in India between the 15th and 12th century B.C., which depicted a cyclic universe expanding and collapsing infinitely many times. Some centuries later, the Greek philosophers questioned themselves about the fundamental constituents of everything (the four elements), and then started to build some primitive models of the cosmos (Anaximander, Anaxagoras, Democritus, Aristotle, Aristarchus), up to the more elaborate Ptolemaic geocentric universe. It was not until the 14th century A.C. that the Polish astronomer Nicolaus Copernicus, based in Aristarchus' model—who already had the right idea, many centuries before, but had been able to convince neither his contemporaries nor future generations—proposed a heliocentric universe, later refined by Johannes Kepler. A milestone was laid when, in 1687, Isaac Newton published his influential *Principia,* where the universe was described as static, steady-state, and infinite.

However, it was not until the second decade of the 20th century that modern cosmology appeared. This happened thanks to the new tools that allowed astronomers to obtain, for the first time in history, the positions and velocities of the celestial bodies (and thus, treat the universe as an ordinary physical system [1–4]), and thanks also to the solid theoretical basis provided by Albert Einstein's general theory of relativity (GR). His first

model, proposed in 1917, was the well-known Einstein's static model [5] where, for consistency, he introduced his now very famous cosmological constant (CC). In the same year, Willem de Sitter [6] proposed another static and closed model describing a universe that was clearly expanding, but it was devoid of any matter or energy, only containing a CC. In spite of being a solution for Einstein's equations, it was generally considered as physically unrealistic. Five years afterwards, in 1922 and 1924, the Russian mathematician Alexander Friedmann obtained families of solutions of the GR equations depicting expanding and contracting universes [7,8]. Later, in 1927, the Belgian scholar Georges Lemaître, using the observational values for speeds and distances graciously provided to him by the reputed astronomers Vesto Slipher and Edwin Hubble, was the first to propose that the universe was actually expanding (however, in this model, it had no origin in time, which extended from minus to plus infinity) [9]. In any case, the value he obtained for the expansion rate was quite close to the one Hubble would obtain two years later (namely, the famous Hubble constant).

Once the evolution of the universe was clear, the question about its origin acquired relevant importance. That the universe had had an origin was proclaimed by Lemaître [10], a few years after his great discovery of the universe's expansion. He even dared to propose a specific model for the beginning of the cosmos, a tremendous explosion of a primeval atom—an idea soon to be scientifically discredited, but that, owing to its extreme simplicity and its beautiful name, the Big Bang, has persisted until now in the popular literature. Later, it was proven that, under very general conditions, the GR equations for the universe imply that it must have started from a "mathematical" singularity (since all GR solutions diverge at some finite value of time in the past), what is now termed the Big Bang (BB) singularity and is at the heart of the so-called Big Bang cosmology (BBC). It is important to clarify from the very beginning that this singularity—although rigorous and unavoidable in principle—is only mathematical (as are most, if not all, singularities appearing in physics). As usually happens, it comes from the simplicity of the theory and appears in a region where the model is no longer valid, e.g., when one assumes that GR holds at *all* scales, what is clearly not true: for it is not valid at small-length scales.

Being more specific, when our patch of universe (the one visible for us) was the size of an atom or less, as-yet-unknown physical effects, such as quantum effects (produced by a quantum field [11] or by holonomy corrections, as in LQC [12]) or non-linear curvature effects in the Hilbert–Einstein action [13,14], did play an important role and could eventually prevent the singularity from occurring.

Moreover, the Big Bang cosmology model had some shortcomings, notably, the horizon and flatness problems, which could finally be overcome thanks to Alan Guth's brilliant idea of *cosmic inflation* [15]. This is only an implementation of the old BBC model, where at very early times, presumably at GUT scales, a very short period of extremely fast accelerated expansion of the universe is assumed to have ocurred. In fact, the inflationary paradigm, devised by Guth at the beginning of the 1980's and improved by Andrei Linde and other cosmologists [16,17], is still at present considered as the simplest viable theory that describes the early universe and that is in agreement with the most recent observations. It also has a good predictive power, because it is able to explain the origin of the present inhomogeneities in the universe as quantum fluctuations during that epoch [18–22], matching well the latest observational data from the Planck survey [23].

A further leap forward occurred when, towards the end of the last century and after very important hints had been accumulated in that direction (see, e.g., [1] for a detailed explanation), it was realized that the universe's expansion is actually accelerating [24,25]. Scientists try to explain this acceleration by introducing a new energy component: *dark energy* [26]. Its most simple realization is the re-introduction of Einstein's cosmological constant, what leads to the so-called $\Lambda$ cold dark matter model (the current standard cosmological model) [1–4]. However, to match with the current observational data, the value of the CC in this model has to be fine-tuned with enormous precision. This is the reason why some cosmologists introduced the idea of *quintessence*, where a scalar field

is responsible for the late-time acceleration, as a new class of dark energy (see [27], for a review of these models).

In this way, one can think of inflation and quintessence as the two different sides of a coin, or, after the introduction by Peebles and Vilenkin, in their seminal paper [28] on the concept of *quintessential inflation* (QI), as simply a unique kind of force which describes both inflation and the accelerated expansion. The idea of a unified picture of the universe, connecting the early and the present accelerated stages, is possible through the introduction of a single scalar field, also named the *inflaton*, as in standard inflation, which at early times produces inflation while at late times provides quintessence.

In addition, the majority of models in QI are so simple that they only depend on two parameters, which are determined by observational data. Since the dynamical behavior during inflation is an attractor, the dynamics of the model are obtained with the value of the scalar field and its derivative—initial conditions—at some moment during this period; this shows the simplicity of quintessential inflation.

The assumption that the universe had an inflationary phase readily implies that a reheating mechanism, to match inflation with the hot Big Bang universe, is needed [15]. This is because the particles existing before the beginning of this period are extremely diluted by the end of inflation, resulting in a very cold universe. The most accepted idea to reheat the universe in the context of QI is through a phase transition, namely, from inflation to kination (a phase where almost all the energy density of inflation turns into a kinetic form [29]), the adiabatic regime being broken, which allows for an abundant production of particles. This mechanism is not unique, since a number of other alternatives can be used. Two of them seem to be the most efficient. The first is the *gravitational particle production* of massless particles, already studied a long time ago in [30–38], and recently applied to quintessential inflation in [28,39] for massless particles, and also the reheating via the gravitational production of heavy massive particles conformally coupled to gravity [40–44]. The second mechanism is called *instant preheating*, which was introduced in [45] and applied for the first time to inflation in [46].

The present review is organized as follows. In Section 2, we discuss the Big Bang model, and obtain first the fundamental equations in an easy way by using a simplified version of the Einstein–Hilbert action, together with the first law of thermodynamics. Once we have these equations, we deal with the Big Bang singularity and with future singularities, and we review some possible ways to remove them, such as in loop quantum cosmology and in semi-classical gravity. Section 3 is devoted to the study of the inflationary paradigm. First of all, the famous horizon and flatness problems are described to then introduce Alan Guth's proposal of cosmic inflation, which solves the shortcomings of Big Bang cosmology. We describe with great detail the slow-roll regime, explaining in a clear way the conditions to obtain it, and its attractor nature. In Section 4, we consider the current cosmic acceleration, starting with Einstein's static model of 1917, where he introduced his celebrated cosmological constant. This constant is now at the heart of the famous Λ cold dark matter model, which is presently the standard cosmological model used by many scientists to depict our universe. The importance of dark matter on a cosmological level, e.g., for the formation of large-scale structures in the universe, cannot be underestimated. It can actually provide a better picture of the current state of the art of the Big Bang cosmological model (for easy-to-follow references, see [47–49]).

However, in such a model, the cosmological constant has to be very finely tuned, which is conceptually a serious problem. This is the reason why other forms of *dark energy* have been introduced, in order to deal with this issue. One of these proposals is *quintessence*, which we review in the context of quintessential inflation: a theory that aims at unifying the early and the late-time accelerated phases of our universe. The reheating of the universe is considered in Section 5, where by using the quantum harmonic oscillator model, we introduce the well-known diagonalization method, with the aim to calculate the energy density of particles created via gravitational particle production. As an application, we calculate the reheating temperature in the so-called *α*-attractor models, which are derived

rather naturally from fundamental super-gravity theories. Finally, in Section 7, some relevant historical notes are included, which describe a few crucial moments and important developments in the history of modern cosmology. A much more detailed account of these historical issues is provided in the recent book by one of the authors [1], which is a perfect complement to the present review. Here, a much more specific, technically solid, and detailed quantitative explanation of the concepts will be given, with all the relevant formulas and with the help of many comprehensible examples.

*Conventions*

Throughout the work, natural units are used, $\hbar = c = k_b = 1$, where $\hbar$ is the reduced Planck constant, $c$ is the velocity of light, and $k_b$ is Boltzmann's constant. In *natural* units, one has:

$$[energy] = [mass] = [temperature] = [length]^{-1} = [time]^{-1}.$$

In these units, the reduced Planck mass reads $M_{pl} = \sqrt{\frac{1}{8\pi G}} \cong 2.44 \times 10^{18}$ GeV, with $G$ being Newton's constant.

Planck's scale:

1. Planck's lenght: $l_{pl} = \left(\frac{G\hbar}{c^3}\right)^{1/2} = 1.616 \times 10^{-33}$ cm. Compare with Bohr's radius: $r_B \cong 5.3 \times 10^{-9}$ cm;

2. Planck's time: $t_{pl} = \frac{l_{pl}}{c} = 5.391 \times 10^{-44}$ s;

3. Planck's mass: $m_{pl} = \left(\frac{\hbar c}{G}\right)^{1/2} = 2.117 \times 10^{-5}$ g. Compare with the proton mass: $m \cong 1.7 \times 10^{-24}$ g;

4. Planck's temperature: $T_{pl} = \frac{m_{pl} c^2}{k_B} \cong 1.4 \times 10^{32}$ K. The current temperature of our universe is 2.73 K. The temperature of the solar surface is approximately 6000 K;

5. Planck's energy density $\rho_{pl} = \frac{m_{pl} c^2}{l_{pl}^3} = 4.643 \times 10^{114}$ erg·cm$^{-3}$. The energy, mass, and temperature are either given in GeV or in terms of $M_{pl}$. For example, the temperature of the universe one second after the Big Bang is around $10^{-3}$ GeV or $10^{-21} M_{pl}$, which, in IS units, is approximately $10^8$ K.

## 2. Big Bang Cosmology

Modern cosmology is based on Einstein's equations (EE) of general relativity (GR), which relates the geometry of spacetime—a four-dimensional manifold—with the matter/energy it contains in the following way: *"matter tells spacetime how to curve and spacetime tells test particles how to move"*. A test particle moves in spacetime along a geodesic curve.

Being more precise, given a distribution of masses, the unknown variables are the coefficients of the metric $g_{\mu\nu}$, and the equations of GR are second-order partial differential equations (PDE) containing some constraints (which are equations of order one or zero). The most important of them is the Hamiltonian constraint: the total Hamiltonian of the system (the universe) vanishes. These equations are:

$$R_{\mu\nu} - \frac{1}{2} R g_{\mu\nu} = \frac{8\pi G}{c^4} T_{\mu\nu}, \qquad \mu, \nu = 0, 1, 2, 3, \tag{1}$$

where $R_{\mu\nu} = R^{\alpha}_{\mu\alpha\nu}$ is the Ricci tensor, $R = R^{\mu}_{\mu}$ is the Ricci scalar (or scalar curvature), and $T_{\mu\nu}$ is the stress–energy tensor (containing all the information about the masses and energies of the universe).

Here,

$$R^{\rho}_{\sigma\mu\nu} = \partial_\mu \Gamma^{\rho}_{\nu\sigma} - \partial_\nu \Gamma^{\rho}_{\mu\sigma} + \Gamma^{\rho}_{\mu\lambda} \Gamma^{\lambda}_{\nu\sigma} - \Gamma^{\rho}_{\nu\lambda} \Gamma^{\lambda}_{\mu\sigma}, \tag{2}$$

is the Riemann curvature tensor, where:

$$\Gamma^{\rho}_{\mu\nu} = \frac{1}{2} g^{\mu\lambda} \left( \partial_{\nu} g_{\lambda\mu} + \partial_{\mu} g_{\lambda\nu} - \partial_{\lambda} g_{\mu\nu} \right), \tag{3}$$

are the Christoffel symbols.

Fortunately, in cosmology, these equations simplify very much. In fact, we just know about our patch of the universe, that is, the part of the cosmos that is visible to us. (Recall that, owing to inflation and also to the finite velocity of light, the whole universe is not visible to us.) The extension of our patch is of the order of 3000 Mpc (1 Megapasec (Mpc) = $3.26 \times 10^6$ light years = $3.08 \times 10^{19}$ km), and it is observed to be very homogeneous (we see the same properties at all places of our patch) and isotropic (it does not matter which direction our telescope is pointing in, properties also remain the same), on scales larger than, say, 100 Mpc, but it does exhibit inhomogeneous structures on smaller scales (as, e.g., on scales of the Milky Way).

So, working on large scales, we can safely assume a homogeneous and isotropic universe, which enormously simplifies the corresponding EE: the only variable is now the cosmic time (spatial coordinates do not appear in the equations due to homogeneity). Thus, we are left with a set of ordinary differential equations (ODE), instead of the original PDE. The hypothesis of homogeneity and isotropy are extremely well supported by surveys of galaxies (e.g., SDSS) and by the recent results of the WMAP and the Planck satellites (for updated information, the reader can look at [50–52]).

### 2.1. Hubble's Law

One of the most important discoveries in human history is the fact that our universe is expanding. This conclusion was reached through the cosmic Doppler effect: light of distant celestial objects is red-shifted, i.e., owing to the universe's expansion, the wavelength of the light emitted by these objects grows (the waves decompress), so the light we see from far-away galaxies is displaced towards the red region of the spectrum. On the contrary, in a hypothetically contracting universe, the wavelength of the emitted light would be compressed, and we would see a displacement towards the blue region of the spectrum. This is what actually happened with the very first measurement performed by Vesto Slipher, in 1912, corresponding to the nearby Andromeda galaxy. It was a blueshift, due to the fact that, in this case, gravitational attraction is much larger than the effect of the local expansion of the universe. Moreover, in fact, Andromeda is approaching the Milky Way at a very high speed: the two galaxies will collide in the future. This is an important lesson we need to learn: cosmic expansion always competes with the gravitational forces of highly massive objects, which provide contributions to the total redshift that astronomers have to disentangle (which, in general, is quite difficult to do). You can see a very nice explanation of the Doppler effect, in the context of GR, in episode 8 of Carl Sagan's "Cosmos" series [53].

One of the fundamental principles of modern cosmology is Hubble's empirical law (1929) [54], which he obtained by measuring (mainly by using Henrietta Leavitt's law for Cepheid variable stars) the distances to spiral nebulae (given in Mpc) and comparing them with the table of speeds (measured in kms) obtained by Vesto Slipher a few years before and, by then, published already in Eddington's aclaimed book. Actually, Hubble's law had been obtained by Georges Lemaître two years before, a fact that was recently recognized, at last [1–4]. As a result, the expansion law is now officially called the Hubble–Lemaître law, which relates the relative velocities of observers as follows: In an expanding, homogeneous, and isotropic universe, the velocity of the observer *B* with respect to *A* is:

$$\vec{v}_{AB} = H(t)\vec{r}_{AB} \Longleftrightarrow \dot{\vec{r}}_{AB} = H(t)\vec{r}_{AB}, \tag{4}$$

where $H(t) > 0$ is the so-called Hubble rate and $\vec{r}_{AB}$ is the vector pointing from *A* towards *B*.

Integrating this equation, one obtains: $\vec{r}_{AB}(t) = a(t)\vec{r}_{AB}(t_i)$, where the dimensionless function $a(t) = e^{\int_{t_i}^{t} H(s)ds}$ is the so-called *scale factor*. What is important is to note that $d_{AB}(t_i) = |\vec{r}_{AB}(t_i)|$ is the distance, at time $t_i$, between points $A$ and $B$, and that at time $t$, this distance is $d_{AB}(t) > d_{AB}(t_i)$ (for $t > t_i$, because $H(t) > 0 \implies a(t) > 1$); thus, the universe is expanding.

In fact, the distance between two points at different epochs is given by the formula $d_{AB}(t_2) = \frac{a(t_2)}{a(t_1)} d_{AB}(t_1)$; that is, the scale factor tells us how the distance between two points scales with time. For a clear idea of this expansion, we can "imagine", as the simplest model, that the universe is an inflating balloon—in more rigorous mathematical terms, the three-dimensional spherical surface of a four-dimensional ball, with time being the radial coordinate—and $r_i$ is its radius at time $t_i$; then, the radius of the balloon at time $t$ would be $a(t)r_i$. In addition, one has $H(t) = \frac{\dot{a}(t)}{a(t)}$, which means that this quantity is the expansion rate of the universe, where reliable observational data tell us that its current value, $H_0$, is approximately 70 $\frac{\text{km/s}}{\text{Mpc}}$.

Before closing this point, we have to say that $H_0$ is both the most important cosmological parameter and also the most difficult to calculate—both because of the difficulty of measuring cosmological distances reliably, and of disentangling the cosmological contribution from other components of the observed redshift of a very distant celestial object object. The calculated value of $H_0$ has changed a lot, from the time of the first reported measurements by Lemaître (600) and Hubble (500), down later to a value of 50, the preferred one some decades ago [55,56]. Even presently, there is a new and quite sharp controversy among different groups (see, e.g., [57] and references therein) concerning what presently sets this value between 67 and 74, according to the results reported by different groups (there is, namely, over a 10% discrepancy, even to this day) [1–4]. This issue has been named the expansion rate tension. This is a very important issue at present, and we would like to provide some more information regarding the nature of this tension, which affects mainly the late-time versus the early time data results. To give a detailed account of this issue lies beyond the scope of the present paper, but a very well-documented version of it can be found, e.g., in [58,59].

### 2.2. The Cosmic Equations

The main ingredient to obtain the dynamical equations of the cosmos at a large scale is the scalar curvature or Ricci scalar, which, for a spatially flat space-time (later, we will see that our universe could be spatially closed, flat, or open), is given by $R(t) = 6(\dot{H}(t) + 2H^2(t))$. In addition, we assume that, at large scales, galaxies can be taken as particles of a homogeneous fluid filling the universe, whose energy density is denoted by $\rho(t)$. The corresponding Lagrangian, in natural units, was given by Hilbert in 1915, immediately after Einstein had postulated, "a la Newton", the equations of GR (see the historical note at the end):

$$\mathcal{L}(t) = V(t)\left(\frac{R(t)M_{pl}^2}{2} - \rho(t)\right), \tag{5}$$

where $V(t) \equiv a^3(t)$ is the *volume* and $M_{pl}$ is the reduced Planck mass.

The corresponding Hilbert–Einstein action reads $S(t) = \int_{t_i}^{t} \mathcal{L}(s)ds$. Then, since:

$$\mathcal{L} = -\frac{\dot{V}^2 M_{pl}^2}{3V} + \ddot{V}M_{pl}^2 - \rho V, \tag{6}$$

and $\ddot{V}M_{pl}^2 = \frac{d}{dt}\left(\dot{V}M_{pl}^2\right)$ is a total derivative, it can be disregarded because it does not have any influence in the dynamical equations. Thus, we will use the following Lagrangian, which only contains first-order derivatives with respect to the volume variable:

$$\bar{\mathcal{L}} = -\frac{\dot{V}^2 M_{pl}^2}{3V} - \rho V = -3H^2 M_{pl}^2 V - \rho V. \tag{7}$$

**Remark 1.** *At this point, we should recall that, in classical mechanics, given a Lagrangian of the form $\mathcal{L}(x, \dot{x}, t)$, performing the variation of the action $S(t) = \int_{t_i}^{t} \mathcal{L}(x(s), \dot{x}(s), s)ds$ with respect to the dynamical variable $x$, one obtains the so-called Euler–Lagrange equation:*

$$\frac{d}{dt}\left(\frac{\partial \mathcal{L}}{\partial \dot{x}}\right) = \frac{\partial \mathcal{L}}{\partial x}, \tag{8}$$

*whose equivalent formulation is to consider the Hamiltonian, obtained via the Legendre transformation:*

$$\mathcal{H} = \dot{x} p_x - \mathcal{L}, \qquad where \qquad p_x \equiv \frac{\partial \mathcal{L}}{\partial \dot{x}}. \tag{9}$$

*Then, the Euler–Lagrange equations are equivalent to the Hamiltonian equations:*

$$\dot{x} = \partial_{p_x}\mathcal{H}, \qquad \dot{p}_x = -\partial_x \mathcal{H}. \tag{10}$$

Coming back to our Lagrangian, the Legendre transformation leads to:

$$\mathcal{H} = \dot{V} p_V - \bar{\mathcal{L}}, \tag{11}$$

where the corresponding momentum is $p_V \equiv \frac{\partial \bar{\mathcal{L}}}{\partial \dot{V}} = -\frac{2\dot{V} M_{pl}^2}{3V}$. Then, a simple calculation shows that the Hamiltonian is given by:

$$\mathcal{H} = -\frac{\dot{V}^2 M_{pl}2}{3V} + \rho V = -3H^2 M_{pl}^2 V + \rho V. \tag{12}$$

As we have already explained, the EE contain some constraints, and one of them states that the Hamiltonian vanishes. So, we have:

$$H^2 = \frac{\rho}{3M_{pl}^2}, \tag{13}$$

which is the well-known Friedmann equation (FE).

**Remark 2.** *The Friedmann equation could also be obtained using the time $Nds = dt$, where $N$ is the so-called lapse function. From this new time, the Lagrangian becomes:*

$$\bar{\mathcal{L}} = -3\frac{\tilde{H}^2}{N} M_{pl}^2 V - \rho V N, \tag{14}$$

*where $\tilde{H} = \frac{1}{a}\frac{da}{ds} = NH$.*
*Then, the variation with respect to the lapse function yields:*

$$\frac{\partial \mathcal{L}}{\partial N} = 0 \implies H^2 = \frac{\tilde{H}^2}{N^2} = \frac{\rho}{3M_{pl}^2}. \tag{15}$$

Once we have obtained the constraint, we need to find the dynamic equation. To do that, we need *the first law of thermodynamics*. Assuming an adiabatic evolution of the universe, i.e., that the universe's entropy is conserved, one has:

$$d(\rho V) = -PdV, \tag{16}$$

where $\rho$ is, once again, the energy density of the fluid and $P$ is its pressure. Taking the derivative with respect to the cosmic time $t$, it can be expressed as a conservation equation (CE):

$$\dot{\rho} = -3H(\rho + P). \tag{17}$$

Next, taking the derivative of the FE and using the CE, one obtains the Raychaudhuri equation (RE):

$$\dot{H} = -\frac{1}{2M_{pl}^2}(\rho + P). \tag{18}$$

Finally, to obtain the dynamics of the universe we need the relation between the pressure and the energy density, i.e., the equation of state (EoS), which, for the moment, we assume has the simple form $P = P(\rho)$.

**Remark 3.** *We should note that the Raychaudhuri equation can also be obtained from the Euler–Lagrange one. In fact, a simple calculation leads to:*

$$\frac{d}{dt}(\partial_{\dot{V}}\bar{\mathcal{L}}) = -2\dot{H}M_{pl}^2. \tag{19}$$

*On the other hand,*

$$\partial_V \bar{\mathcal{L}} = 3H^2 M_{pl}^2 - \partial_V(\rho V); \tag{20}$$

*thus, using the Friedmann equation and the first law of thermodynamics, one obtains:*

$$\partial_V \bar{\mathcal{L}} = \rho + P, \tag{21}$$

*which leads to the Raychaudhuri equation.*

Summing up, *the constituent equations* in cosmology are:

$$H^2 = \frac{\rho}{3M_{pl}^2} \quad \text{Friedmann equation.}$$

$$\dot{\rho} = -3H(\rho + P) \quad \text{conservation equation.}$$

$$\dot{H} = -\frac{1}{2M_{pl}^2}(\rho + P) \quad \text{Raychaudhuri equation.}$$

$$P = P(\rho) \quad \text{equation of state.}$$

Note that the variables are $H$ and $\rho$, but they are related via the FE. So, in practice, we have only one unknown variable. Moreover, the CE and the RE are equivalent, which means that we only need to solve one of them. For example, the CE, which can be written as:

$$\dot{\rho} = -\frac{\sqrt{3\rho}}{M_{pl}}(\rho + P(\rho)), \tag{22}$$

is only a simple and solvable first-order differential equation.

Next, we define the EoS parameter as $w_{eff} = \frac{P}{\rho}$, which, using the constituent equations, can be written as follows:

$$w_{eff} = -1 - \frac{2\dot{H}}{3H^2}. \tag{23}$$

Then, from the EoS parameter, the *acceleration equation* could be written as follows:

$$\frac{\ddot{a}}{a} = \dot{H} + H^2 = -\frac{H^2}{2}(1 + 3w_{eff}), \tag{24}$$

and thus, one can conclude that the universe is decelerating for $w_{eff} > -1/3$, and that it is accelerating for $w_{eff} < -1/3$. In addition, from the Raychaudhuri equation, for $w_{eff} > -1$, the Hubble parameter decreases, and for $w_{eff} < -1$ (phantom fluid), it increases.

Finally, from the first law of thermodynamics, one can easily show that for $w_{eff} = w = $ constant, the energy density scales as $\rho \propto a^{-3(w+1)}$.

*2.3. Singularities*

We consider the following linear EoS $P = w\rho$, where $w > -1$ (non-phantom fluid) is a constant EoS parameter ($w = 0$ for a dust fluid and $w = 1/3$ for radiation). The combination of the FE and the RE leads to:

$$\dot{H} = -\frac{3}{2}(1 + w)H^2, \tag{25}$$

whose solution is given by:

$$H(t) = \frac{H_0}{\frac{3}{2}(1 + w)H_0(t - t_0) + 1} = \frac{2}{3(1 + w)(t - t_s)}, \tag{26}$$

where $t_0$ is the present time, $H_0$ is the current value of the Hubble rate, and $t_s = t_0 - \frac{2}{3(1+w)H_0}$ is the time when the singularity appears. Inserting this expression in the FE, one obtains:

$$\rho(t) = \frac{4M_{pl}^2}{3(1 + w)^2(t - t_s)^2}. \tag{27}$$

So, we see that the solutions $H(t)$ and $\rho(t)$ diverge at time $t_s < t_0$. This is the so-called Big Bang (BB) singularity (see the historical note at the end to understand where that name comes from), where the scale factor:

$$a(t_s) = \lim_{t \to t_s} a(t_0)e^{-\int_t^{t_0} H(s)ds} = 0; \tag{28}$$

that is, the "radius" of the universe vanishes at the BB singularity.

On the other hand, the age of our universe is given approximately by:

$$t_0 - t_s = \frac{2}{3(1 + w)H_0} \cong 14 \text{ billion years}, \tag{29}$$

where the value of the Hubble rate that we have used currently is $H_0 \cong 70 \frac{\text{km/s}}{\text{Mpc}}$ and $w \sim 0$.

Here, it is important to remark that there are other kind of mathematical singularities. In fact, when the EoS is non-linear, as in $P = -\rho + A\rho^\alpha$, one can encounter these different future singularities [60]:

1.   Type I (Big Rip): For $t \to t_s$, $a \to \infty$, $\rho \to \infty$, and $|P| \to \infty$;
2.   Type II (Sudden): For $t \to t_s$, $a \to a_s$, $\rho \to \rho_s$, and $|P| \to \infty$;

3.　Type III (Big Freeze): For $t \to t_s$, $a \to a_s$, $\rho \to \infty$, and $|P| \to \infty$;
4.　Type IV (Generalized Sudden): For $t \to t_s$, $a \to a_s$, $\rho \to 0$, $|P| \to 0$, and higher-order derivatives of $H$ diverge.

### 2.3.1. Big Rip Singularity

This singularity is the future equivalent to the Big Bang one, and it is obtained, for instance, when one deals with a phantom fluid with a linear EoS ($w < -1$). The solution is given by (26), but now $t_s > t_0$; that is, the singularity appears at late times.

### 2.3.2. Sudden Singularity

In [61], Barrow proposed a new kind of finite-time future singularity appearing in an expanding FLRW universe. The singularity may also show up without violating the strong energy conditions $\rho > 0$ and $3P + \rho > 0$. This singularity was named the *sudden singularity*.

To deal with these kind of singularities, we consider a nonlinear EoS $P = -\rho - f(\rho)$. In this case, the conservation equation becomes $\dot{\rho} = 3H f(\rho)$, and using the Friedmann equation, one obtains:

$$\dot{\rho} = \sqrt{3}\frac{\rho^{1/2}}{M_{pl}}f(\rho). \tag{30}$$

Choosing, as in [62], $f(\rho) = \frac{AM_{pl}}{\sqrt{3}}\rho^{\nu+\frac{1}{2}}$, where $A$ and $\alpha$ are two parameters, one obtains the first-order differential equation:

$$\dot{\rho} = A\rho^{\nu+1}, \tag{31}$$

whose solution is:

$$\rho(t) = \begin{cases} (\rho_0^{-\nu} - \nu A(t - t_0))^{-1/\nu} & \nu \neq 0 \\ \rho_0 e^{A(t-t_0)} & \nu = 0, \end{cases} \tag{32}$$

where $\rho_0$ is the current value of the energy density.

Now, we consider the case $\nu < -1/2$. The Hubble rate is given by:

$$H(t) = \frac{1}{\sqrt{3}M_{pl}}\left(\rho_0^{-\nu} - \nu A(t - t_0)\right)^{-1/2\nu}, \tag{33}$$

which, introducing the time $t_s = t_0 + \frac{\rho_0^{-\nu}}{\nu A}$, can be written as:

$$H(t) = \frac{1}{\sqrt{3}M_{pl}}\left(\nu A(t_s - t)\right)^{-1/2\nu}, \tag{34}$$

and thus, the scale factor is given by:

$$\ln\left(\frac{a(t)}{a_s}\right) = -\frac{2}{3M_{pl}A(2\nu - 1)}\left(\nu A(t_s - t)\right)^{(2\nu-1)/2\nu}. \tag{35}$$

Then, for a non-phantom fluid, namely, for $A < 0$ (the effective EoS parameter is $w_{eff} \equiv \frac{P}{\rho} = -1 - \frac{AM_{pl}}{\sqrt{3}}\rho^{\nu-\frac{1}{2}} > -1$), we see that $t_s > t_0$, and since $\rho(t_s)$ vanishes, we find that the pressure $P$ diverges at the instant $t_s$, yielding a future Sudden singularity.

### 2.3.3. Big Freeze Singularity

For the same EoS as in the previous subsection, here, we consider the case $\nu > 1/2$ and $A > 0$, that is, a phantom fluid, which implies that $t_s > t_0$. From (35), we see that $a(t) \to a_s$ when $t \to t_s$, and from (33), we deduce that both the energy density and the pressure diverge at that instant, which leads to a Big Freeze singularity.

2.3.4. Generalized Sudden Singularity

We continue with the same EoS, but with $-1/2 < \nu < 0$ and $A < 0$ (non-phantom fluid).

In this situation, $t_s > t_0$ and, once again, the scale factor converges to $a_s$ when $t \to t_s$; now, however, the energy density and the pressure go to zero as the cosmic time approaches $t_s$. In addition, looking at the Hubble rate obtained in Formula (33), we easily conclude that when $-\frac{1}{2\nu}$ is not a natural number, then some high-order derivative of the Hubble parameter will diverge at $t = t_s$, thus obtaining a Generalized Sudden singularity.

To end this section, note that the case $0 < \nu < 1/2$ and $A < 0$ (non-phantom fluid) corresponds to a Big Bang singularity and, for $0 < \nu < 1/2$ and $A > 0$ (phantom fluid), one recieves a Big Rip singularity. In addition, when $\nu = 0$ and $A > 0$, one obtains the so-called Little Rip [63,64], where $w < -1$ and asymptotically converges to $-1$. In fact, in this case, the energy density is given by $\rho(t) = \rho_0 e^{A(t-t_0)}$, which diverges when $t \to \infty$. So,

$$w = \frac{P}{\rho} = -1 - \frac{AM_{pl}}{\sqrt{3\rho}} \to -1. \tag{36}$$

Finally, it remains the case $\nu = 1/2$ and $A > 0$, where the Hubble rate is given by:

$$H = \frac{2}{\sqrt{3}M_{pl}A(t_s - t)}, \tag{37}$$

and thus, the scale factor is given by:

$$\ln\left(\frac{a(t)}{a_0}\right) = -\frac{2}{\sqrt{3}M_{pl}A} \ln\left(\frac{t_s - t}{t_s - t_0}\right), \tag{38}$$

which means that the scale factor diverges when $t \to t_s$, and thus, we obtain a Big Rip singularity.

*2.4. Removing Singularities*

It is very important to realize that the singular solutions we have found are just mathematical solutions of the EE. We know that GR is a viable theory that has been proven to match the observational data at low energy densities, but we still do not know what the valid physical laws are at very high energies. Recall that, at Planck scales, $\rho_{pl} \sim M_{pl}^4 \sim 10^{114}$ erg/cm$^3$ and $l_{pl} \sim M_{pl}^{-1} \sim 10^{-33}$ cm.

We should take this into account, and also the very clear fact that the EE—as it happens with Newton's ones—are of no use to describe small scales. At the nuclear and even the atomic scales, it is already accepted by the physics community that we need to quantize gravity in order to depict our universe at very early times, at least up to Planck scales— beyond that scale, no physical theory has been proven right, up to now. However, for the moment, nobody knows how to obtain a quantum theory of gravity; it might even be simply impossible. Maybe gravity is a force of a very different nature, as compared with the electromagnetic and the nuclear forces. It could even be non-fundamental, but rather a kind of emergent phenomena not to be described with gauging and quantization.

In short, adhering to Einstein's viewpoint, we should not worry too much about these mathematical singularities, since nature, and true physical descriptions of it, are always free of them.

In spite of all these problems, some scientific communities attempt to obtain physical (or, if you want, metaphysical, until they can be validated with actual experiments) laws that are valid at high energy densities in several different ways:

1.  Introducing quantum effects (more precisely, holonomy corrections), which could be disregarded at low energy densities, as in loop quantum cosmology (LQC). These

quantum effects produce a modification of the FE (the holonomy-corrected FE), which becomes [65–68]:

$$H^2 = \frac{\rho}{3M_{pl}^2}\left(1 - \frac{\rho}{\rho_c}\right),$$

(39)

where $\rho_c \cong 0.4\rho_{pl}$ is the so-called *critical density*.

Note that now, the modified FE depicts an ellipse in the plane $(H, \rho)$ (recall that, in GR, the FE depicts a parabola), which is a bounded curve, so the energy density is always finite. In fact, it is always bounded by the critical one. As a consequence, the singularities such as the BB or the Type I and III do not exist in LQC. About the BB singularity, people say that, in LQC, the Big Bang is replaced by a Big Bounce (the universe bounces from the contracting to the expanding phase).

In addition, for low energy densities $\rho \ll \rho_c$, the FE in GR (the usual one) is recovered. Thus, singularities of Type II and IV also exist in LQC. Finally, for a non-phantom fluid, the movement throughout the ellipse is clockwise; that is, the universe starts in the contracting phase, with an infinite size, and bounces to enter the expanding one.

2. Introducing non-linear effects in the Ricci scalar. We have already seen that, in GR, the Hilbert–Einstein Lagrangian is $\mathcal{L} = V\left(\frac{RM_{pl}^2}{2} - \rho\right)$. Then, the idea is that at high energy densities the Ricci scalar is replaced by a more general function $R \to F(R)$, which satisfies $F(R) \to R$ when $R \to 0$, in order to recover GR at low energy densities (recall that $R = 6(\dot{H} + 2H^2) \sim \rho / M_{pl}^2$) [13,69–71].

To obtain the Hamiltonian of the system, one has to use the so-called Ostrograski construction, because the Lagrangian contains second derivatives of $V$. The Hamiltonian constrain leads to the following modified FE in $F(R)$ gravity:

$$-6F''(R)\dot{R}H + F'(R)(R - 6H^2) - F(R) + \frac{2\rho}{M_{pl}^2} = 0.$$

(40)

This is a second-order differential equation with respect to the Hubble rate, which, together with the CE and the EoS, give the dynamics of the universe. However, contrary to the constituent equations in GR, one needs to perform numeric calculations in order to solve the equations in $F(R)$ gravity. The most famous model is $R^2$ gravity (sometimes known as the Starobinsky model) given by $F(R) = R + \alpha R^2$, which was extensively studied in the Russian literature.

3. One can also introduce quantum effects produced by massless fields conformally coupled with gravity, obtaining the so-called *semiclassical gravity*. If one considers some massless fields conformally coupled with gravity, the vacuum stress tensor acquires an anomalous trace, given by [72]:

$$T_{vac} = \alpha\partial_\mu\partial^\mu R - \frac{\beta}{2}G,$$

(41)

where $R$ is, once again, the Ricci scalar, and $G = -2(R_{\mu\nu}R^{\mu\nu} - \frac{1}{3}R^2)$ is the Gauss–Bonnet invariant. In terms of the Hubble parameter, one has:

$$T_{vac} = 6\alpha\left(\frac{d^3H}{dt^3} + 12H^2\dot{H} + 7H\ddot{H} + 4\dot{H}^2\right) - 12\beta(H^4 + H^2\dot{H}).$$

(42)

The coefficients $\alpha$ and $\beta$ are fixed by the regularization process. For instance, using adiabatic regularization, one obtains [11]:

$$\alpha = \frac{1}{2880\pi^2}\left(N_0 + 6N_{1/2} + 12N_1\right) > 0,$$

$$\beta = \frac{-1}{2880\pi^2}\left(N_0 + \frac{11}{2}N_{1/2} + 62N_1\right) < 0,$$

(43)

while point splitting yields [72]:

$$\alpha = \frac{1}{2880\pi^2}(N_0 + 3N_{1/2} - 18N_1),$$

$$\beta = \frac{-1}{2880\pi^2}(N_0 + \frac{11}{2}N_{1/2} + 62N_1),$$ (44)

where $N_0$ is the number of scalar fields, $N_{1/2}$ is that of four-component neutrinos, and $N_1$ is the number of electromagnetic fields.

Here, it is important to note, as pointed out in [73], that the coefficient $\alpha$ is arbitrary, although it is influenced by the regularization method and also by the fields present in the universe, but $\beta$ is independent of the regularization scheme and it is always negative.

Now, we are interested in the value of the vacuum energy density, namely, $\rho_{vac}$. Since the trace is given by $T_{vac} = \rho_{vac} - 3P_{vac}$, inserting this expression in the conservation equation $\dot{\rho}_{vac} + 3H(\rho_{vac} + P_{vac}) = 0$, one obtains:

$$\dot{\rho}_{vac} + 4H\rho_{vac} - HT_{vac} = 0,$$ (45)

a first-order linear differential equation, which can be integrated by using the method of variation of constants, leading to:

$$\rho_{vac} = 6\alpha\left(3H^2\dot{H} + H\ddot{H} - \frac{1}{2}\dot{H}^2\right) - 3\beta H^4 + Ca^{-4},$$ (46)

where $C$ is an integration constant, which vanishes for flat space-time. This can be understood as follows: for a static space-time, $\rho_{vac}$ reduces to $Ca^{-4}$, and the flat space-time reduces to Minkowski, for which $\rho_{vac} = 0$, and thus, $C = 0$. Therefore, in semi-classical gravity, the Friedmann equation becomes:

$$H^2 = \frac{\rho + \rho_{vac}}{3M_{pl}^2}.$$ (47)

Here, we will consider the empty flat case, which corresponds to $\rho = 0$ and $C = 0$. There, since $\beta < 0$, one has a de Sitter solution $H_+ = \sqrt{-\frac{1}{3\beta}}$. In addition, the Friedmann equation is, in the empty case:

$$3M_{pl}^2 H^2 = 6\alpha\left(3H^2\dot{H} + H\ddot{H} - \frac{1}{2}\dot{H}^2\right) - 3\beta H^4,$$ (48)

and, for $H = 0$, becomes:

$$\ddot{H} = \frac{\dot{H}^2}{2H},$$ (49)

whose solution is given by $H(t) = at^2$; this shows that, for this model, the branches $H > 0$ and $H < 0$ decouple, i.e., the universe cannot transition from the expanding to the contracting phase, and vice versa.

Next, performing the change of variable $p = \sqrt{H}$ (we are here considering that the universe expands), the semi-classical Friedmann equation becomes [74]:

$$\frac{d}{dt}\left(\dot{p}^2/2 + V(p)\right) = -p^2\dot{p}^2,$$ (50)

where:

$$V(p) = -\frac{p^2}{24\alpha}\left(1 + \beta p^4\right).$$ (51)

The corresponding dynamical system can be written as:

$$\begin{cases} \dot{p} = & y \\ \dot{y} = & -3p^2 y - V'(p), \end{cases}$$

(52)

which can be viewed simply as the dynamics, with friction, of a particle under the action of a potential.

There are two different situations (we use the notation $p_+ = \sqrt{H_+}$):

(a)    Case $\alpha > 0, \beta < 0$. Here, the system has two fixed points: $(0,0)$ is an unstable critical point, and $(p_+, 0)$ is stable (it is the minimum of the potential). Solutions are only singular at early times. At late times, they oscillate and shrink around a stable point, that is, $(p_+, 0)$ is a global attractor. In addition, there is a solution that ends at $(0,0)$, and only one nonsingular solution that starts at $(0,0)$ (with zero energy) and ends at $(p_+, 0)$;

(b)    Case $\alpha < 0, \beta < 0$. This is the famous Starobinsky model [75]. The system has two critical points: $(0,0)$ is a stable critical point, and $(p_+, 0)$ is a saddle point (it is the maximum of the potential). There are solutions that do not cross the axis $p = p_+$; these solutions are singular at early and late times: they correspond to the trajectories that cannot pass the top of the potential. There are other solutions that cross the axis $p = p_+$ twice; they are also singular at early and late times. These trajectories pass the top of the potential, bounce at $p = 0$, and pass once again the top of the potential. There are solutions that cross the axis $p = p_+$ once. These solutions are singular at early times; however, at late times, the solutions spiral and shrink to the origin. These solutions pass the top of the potential once, and then bounce some number of times about $p = 0$, shrinking to $p = 0$. Finally, there are only two unstable non-singular solutions: one goes from $(p_+, 0)$ to $(0,0)$, and the other is the de Sitter solution $(p_+, 0)$. What seems a little bit strange is that the title of Starobinsky's paper [75] is "A new type of isotropic cosmological models without singularity". In fact, as we have just shown, in that model, there is only one non-singular solution, but it is unstable, and thus, non-physical.

*2.5. Chronology of the Universe*

To explain the different phases of the universe, first of all, we need to recall some basic elements of thermodynamics. For a relativistic fluid (made of light particles with velocities comparable to the speed of light) in thermal equilibrium at temperature $T$, we know that:

1.    The energy density is given by $\rho = \frac{\pi^2}{30} g_* T^4$, where $g_*$ is the number of degrees of freedom, which for the modes in the standard model are $28 + \frac{7}{8} \times 90 = 106.75$;

2.    For a relativistic fluid, pressure is related to energy via the linear relation $P = \rho/3 \iff w_{eff} = 1/3$;

3.    The number density of particles is $n = \frac{\zeta(3)}{\pi^2} g_* T^3$, where $\zeta$ is the Riemann zeta function;

4.    The entropy density is $s = \frac{\rho + P}{T} = \frac{2\pi^2}{45} g_* T^3$.

The total entropy is $S \equiv sa^3$, so for an *adiabatic process*, i.e., when the total entropy is conserved, one has:

$$Ta = \text{ constant} \implies T\frac{a}{a_0} = \text{ constant} \implies \frac{T}{z+1} = \text{ constant},$$

(53)

where $a_0$ is the present value of the scale factor and $z$ is the cosmic redshift.

Since the current temperature of our universe is $T_0 \cong 2.73$ K $\cong 2.3 \times 10^{-13}$ GeV, one has:

$$\frac{T/\text{GeV}}{z+1} = 2.3 \times 10^{-13}.$$

(54)

We also need the relation between the temperature of the universe and its age. For this, we use the the value of the Hubble rate for a radiation-dominated universe $H(t) = \frac{1}{2t}$. From the FE and the Stefan–Boltzmann law, in natural units, we have:

$$\frac{1}{4t^2} = \frac{\pi^2}{90} g_* \frac{T^4}{M_{pl}^2} \implies T = \sqrt{\frac{M_{pl}}{2\pi}} \left(\frac{90}{g_*}\right)^{1/4} \frac{1}{\sqrt{t}}. \tag{55}$$

Next, denoting by $t_{sec}$ the age of the universe in seconds, we have

$$t_{sec} \times 1s \cong 2t_{sec} \times 10^{43} t_{pl} \cong 4t_{sec} \times 10^{42} M_{pl}^{-1}.$$

Inserting this relation in (55), one obtains the following expression for the temperature of the universe (in MeV) in terms of its age (in seconds):

$$T_{MeV} = \frac{\mathcal{O}(1)}{\sqrt{t_{sec}}}. \tag{56}$$

From there, in the BB model the chronology of our universe can be described as follows:

1. Planck scale. $T_{pl} \sim M_{pl} \cong 2.4 \times 10^{21}$ MeV, which means that the Planck scale is reached at $t_{pl} \sim 10^{-43}$ seconds after the BB. It is very important to remark that no reliable physical theory can be invoked before this time. This is a crucial, insurmountable constraint that any serious physicist knows well, but that, too often, is kept hidden under the carpet (nobody seems to be interested in proclaiming the shortcomings of present-day fundamental physics). The corresponding redshift has a value of $z_{pl} \sim 10^{31}$;

2. Grand unification theory (GUT) scale. It enters when the temperature goes down to $T_{GUT} \sim 10^{16}$ GeV, i.e., for $t_{GUT} \sim 10^{-36}$ s or $z_{GUT} \sim 10^{29}$. The three forces of the standard model (electromagnetic, weak, and strong), which constituted a unique force until then, start to become separated forces below this temperature;

3. Electroweak epoch. It occurs at $T_{EW} \sim 10^{15}$–$10^9$ GeV, i.e., when $t_{EW} < 10^{-32}$ s or $z_{EW} < 10^{22}$. The strong interaction clearly decouples from the electroweak one;

4. Radiation-dominated era. It is set up when $T_{rad} \sim 1$ eV, i.e, $t_{rad} \sim 10^{12}$ s or $z_{rad} \sim 4000$. The energy density of nearly massless relativistic particles dominates. From then on, the weak and electromagnetic forces become separated so that, finally, during this period, all the different forces decouple and become distinct. Electromagnetic radiation dominates the energy content of the universe at this epoch;

5. Matter domination era. It occurs for $T_{matt} < 1$ eV. The energy density of matter dominates, at last. Clusters of galaxies and stars start to be formed during this period due to the omnipresent gravitational force that now overcomes radiation pressure;

6. Recombination. This period starts when the temperature goes down to around 3000 K, the redshift being $z_{rec} \sim 1100$. This happens some 300,000 years after the BB. At this epoch, nearly all free electrons and protons recombine and form neutral hydrogen. Photons decouple from matter and can travel freely, for the first time, throughout the whole universe. They originate what is observed today as cosmic microwave background (CMB) radiation (in that sense, the cosmic background radiation is infrared (and some red) black-body radiation emitted when the universe was at a temperature of some 3000 K, redshifted by a factor of 1100 from the visible spectrum to the microwave spectrum);

7. Dark energy era. It starts when the falling temperature reaches $T_{DE} \sim 10^{-3}$ eV (correspondingly, $z_{DE} \sim 4$), that is, around 3 billion years ago, and lasts up to present time, extending into the future. The universe is dominated by an unknown sort of energy, called *dark energy*, and under its influence, it starts to accelerate (the so-called current cosmic acceleration or late-time acceleration).

A final but important observation is that, as we will discuss in the next section, the classic BB model, in spite of being able to describe the first stages of the universe's evolution, had some very serious shortcomings, which could only be overcome by introducing, at a very early time—most likely at GUT scales—a new, very brief stage named *cosmic inflation*, where the volume of our universe inflated by more than 65 e-folds in an incredibly short period of time.

## 3. Inflation

BB cosmology had some serious troubles, the most famous of them being the horizon problem, pointed out for the first time by Wolfgang Rindler, and the flatness issue, clearly described by Robert Dicke.

### 3.1. The Problems of BB Cosmology

1. The horizon problem: Imagine two observers, $A$ and $B$, in a circumference of variable radius $a(t)$ and separated by an angle $d\theta$. The time it takes for a light signal emitted from A to reach B is $dt$. Then, as in natural units the speed of light is 1, we have $a(t)\frac{d\theta}{dt} = 1$; the distance travelled by a signal of light emitted at time $t_i$ and arriving at time $t_f$ is $d(t_i, t_f) = a(t_f) \int_{t_i}^{t_f} \frac{dt}{a(t)}$. This is what happens in an expanding universe. Let $t_0$ be the present time, and we take $t = 0$ as the singularity, i.e., when the BB occurs. Thus, the distance travelled by a signal emitted at time $t = 0$ and received now is

$$d_0 = a_0 \int_0^{t_0} \frac{dt}{a(t)}.$$

   This is the present horizon size (the size of our patch of universe). We cannot see beyond that distance, i.e., we cannot see galaxies which are further away than the present horizon size.

   To simplify, we will assume that the universe is matter-dominated, that is, $H(t) = \frac{2}{3t} \implies a(t) = a_0 \left(\frac{t}{t_0}\right)^{2/3}$, and thus, $d_0 \sim t_0 \sim H_0^{-1}$. We know that our universe is now very homogeneous, so at Planck scales it had to be extremely homogeneous, but at that time the size of our universe was $d_{pl} = \frac{a_{pl}}{H_0 a_0}$. (Recall that, due to the Hubble law, in an expanding universe, $d_{AB}(t_2) = \frac{a(t_2)}{a(t_1)} d_{AB}(t_1)$.)

   This quantity has to be compared with the size of the causal regions (the distance that light travels from the Big Bang to the Planck era), which is $d_c \sim t_{pl} \sim H_{pl}^{-1}$. We calculate the ratio:

$$\frac{d_{pl}}{d_c} \sim \frac{H_{pl} a_{pl}}{H_0 a_0} = \frac{\dot{a}_{pl}}{\dot{a}_0} = \frac{H_{pl} T_0}{H_0 T_{pl}} \sim \frac{T_0}{H_0}, \tag{57}$$

   where we have used the adiabatic evolution of the universe, $aT = $ constant, and that, in natural units, $H_{pl} \sim T_{pl} \sim M_{pl}$.

   Using that, now, $H_0 \sim 6 \times 10^{-61} M_{pl}$ and $T_0 \sim 5 \times 10^{-31} M_{pl}$, we obtain

$$\frac{d_{pl}}{d_c} \sim 10^{30},$$

   which means that, at Planck scales, there are $10^{90}$ disconnected regions. Then, assuming that inhomogeneities cannot be dissolved by ordinary expansion, how is it possible that our present universe be so homogeneous, and that the cosmic microwave background (CMB) radiation has practically the same temperature in all directions? This seems impossible, if it comes from a patch, which at Planck scales, contains so many regions that have never been in causal contact (they have never exchanged information). This is the well-known horizon problem.

An equivalent way to see this problem goes as follows: As we have already explained, the decoupling, or the last scattering, is thought to have occurred at recombination, i.e., about 300,000 years after the Big Bang, or at a redshift of about $z_{rec} \approx 1100$. We can determine both the size of our universe and the physical size of the particle horizon that had existed at this time.

The size of our universe coincides approximately with the size of the last scattering surface, which, currently, is approximately $H_0^{-1}$, so that, at recombination, the diameter of the last scattering surface was $d_{rec} = \frac{a_{rec}}{a_0 H_0}$. At that time, the size of a causally connected region is $d_c \sim H_{rec}^{-1}$. Then, we have:

$$\frac{d_{rec}}{d_c} \sim \frac{H_{rec} a_{rec}}{H_0 a_0} = \frac{H_{rec}}{H_0(1 + z_{rec})}. \tag{58}$$

Next, taking into account that the energy density of matter scales as $\rho_{m,rec} = \rho_{m,0}(1 + z_{rec})^3$, and further, that at recombination the universe is matter-dominated, i.e., $H_{rec} = \sqrt{\frac{\rho_{m,rec}}{3M_{pl}^2}}$, and that at present time $\rho_{m,0} = 3\Omega_{m,0}M_{pl}^2 H_0^2$, with $\Omega_{m,0} \cong 0.3$, one obtains:

$$\frac{d_{rec}}{d_c} \sim \sqrt{\Omega_{m,0}(1 + z_{rec})} \sim 18. \tag{59}$$

As a consequence, in the last scattering surface, there are regions which are causally disconnected; however, it turns out that the CMB has practically the same temperature in all directions.

To simplify, the horizon size is of the order $1/H_0 \sim 14$ billion lightyears, which coincides with the age of the universe $1/H_0 \sim 14$ billion years ago. So, imagine a region that is at a distance of 10 billion lightyears from us, and another region, in the opposite direction, that is at the same distance from us. The question is, how it is possible that both regions, which are about 20 billion lightyears apart, emitted light at the same temperature?

This is an apparent "paradox" in a static or decelerating universe, but as we will see in the next section, the paradox is overcome when one assumes a short superluminal expansion phase at early times.

2.  The flatness problem: Up to now, we have only considered the dynamical equation for flat space, but, in fact, space could have positive or negative curvature. When one considers the general case, the FE becomes:

$$H^2 = \frac{\rho}{3M_{pl}^2} - \frac{\kappa}{a^2}, \tag{60}$$

with $\kappa = -1, 0, 1$ (open, flat, and closed cases, respectively).

This equation can be written as follows:

$$\Omega - 1 = \frac{\kappa}{a^2 H^2},$$

where $\Omega \equiv \frac{\rho}{3H^2 M_{pl}^2}$ is the ratio of the energy density to the *critical one*.

Evaluating at the Planck time and at the present time, one obtains:

$$\frac{|\Omega - 1|_{pl}}{|\Omega - 1|_0} = \frac{H_0^2 a_0^2}{H_{pl}^2 a_{pl}^2} = \frac{\dot{a}_0^2}{\dot{a}_{pl}^2} = \frac{H_0^2 T_{pl}^2}{H_{pl}^2 T_0^2} \sim \frac{H_0^2}{T_0^2} \sim 10^{-60}, \tag{61}$$

where we have used the adiabatic evolution of the universe.

Thus, in order to have $|\Omega - 1|_0 \cong 0.01$ (the present observational data), the value of $\Omega - 1$ at early times has to be fine-tuned to values amazingly close to zero, but

without letting it be exactly zero—this is the flatness problem, which sometimes is also dubbed *the fine-tuning problem*.

To better understand this fine tuning: our very existence depends on the fantastically close balance between the actual density and the critical density in the early universe. If, for instance, the deviation of $\Lambda$ from one at the time of nucleosynthesis had been one part in 30 thousand, instead of one part in 30 trillion, the universe would have collapsed in a Big Crunch after only a few years. In that case, galaxies, stars, and planets would not have had time to form, and cosmologists would never have existed.

*3.2. Inflation: The Basic Idea*

The solution of the horizon, flatness, and the other problems accumulated by Big Bang cosmology over several decades, was obtained by Alan Guth in 1981, in his seminal paper: *"The Inflationary universe: A Possible Solution to the Horizon and Flatness Problems"* (PRD23, pg. 347–356).

The key point to solve these problems is to consider the ratio $\frac{\dot{a}_{pl}}{\dot{a}_0}$ (this quantity appears explicitly in the horizon and flatness problems). If gravity is attractive, then this ratio is necessarily larger than one, because gravity decelerates the expansion. Therefore, the conclusion $\frac{\dot{a}_{pl}}{\dot{a}_0} \gg 1$ can be avoided only if we assume that, during some period in the cosmic expansion, gravity acted as a "repulsive" force, thus accelerating expansion. In this case, we can have $\frac{\dot{a}_{pl}}{\dot{a}_0} < 1$, and the creation of our patch of universe from a single causally connected domain may become possible.

More precisely, let $a_b$ and $a_e$ be, respectively, the values of the scale factor at the beginning and at the end of the accelerated expansion period, which is named *inflation*. Since $H = \frac{\dot{a}}{a}$, integrating this equation, we have $a_e = a_b e^N$, where $N = \int_{t_b}^{t_e} H(t)dt$ is *the number of e-folds that inflation lasts for*. Assuming that, at the beginning of inflation, which occurred at almost the very beginning of the universe, there was a small patch of it causally connected (light had enough time to travel from one to any other point in that domain), then inflation blows up this region to a very large one, preserving the homogeneity as intact while expanding enormously our patch of the universe.

In order that this patch encompasses our whole universe, as we observe it now, we will see that it is necessary that the number $N$ of e-folds the patch increases by has to be higher than 65. During inflation, the volume grows $e^{3N}$ times; (by this proportion, the volume of an atom would turn into that of an orange.)

Going a bit further, to compute the number of e-folds needed and the time for which inflation should last, we will assume that inflation starts and ends at the GUT scale, i.e, when $H_{GUT} \sim 10^{14}$ GeV and $T_{GUT} \sim 10^{16}$ GeV. The size of our universe at the beginning of inflation is $d_{GUT} = \frac{a_b}{H_0 a_0}$, and a causally connected region has size $d_b \sim H_{GUT}^{-1}$. Then,

$$\frac{d_{GUT}}{d_b} \sim \frac{H_{GUT} T_0}{H_0 T_{GUT}} \sim 10^{-2} \frac{T_0}{H_0} \sim 10^{28}.$$

Since, at the end of inflation, the size of a causally connected region is $d_e = \frac{a_e}{a_b} d_b = e^N d_b$, we will have

$$\frac{d_{GUT}}{d_e} \sim e^{-N} 10^{28} \leq 1.$$

Thus,

$$e^N \geq 10^{28} \implies N \geq 28 \ln(10) \sim 65,$$

as already advanced.

In the same way, for the flatness problem, one has:

$$|\Omega - 1|_0 \sim 10^{56} |\Omega - 1|_{GUT}; \tag{62}$$

however, assuming an inflationary period at GUT scales, one has:

$$|\Omega - 1|_e \sim e^{-2N}|\Omega - 1|_b. \tag{63}$$

So, supposing that, prior to inflation, the universe was actually fairly strongly $|\Omega - 1|_b \sim 1$, one has:

$$|\Omega - 1|_0 \sim 10^{56}e^{-2N}|\Omega - 1|_e, \tag{64}$$

and thus, for $N \geq 65$, the flatness problem is solved.

On the other hand, since, during inflation, $H$ is nearly constant (as we will see in the next subsection), we have (approximately) $N \sim H_{GUT}(t_e - t_b)$. Then, for $N \sim 10^2$, one obtains:

$$t_e - t_b \sim 10^{-12}\text{GeV}^{-1} \sim 10^6 M_{pl}^{-1} \sim 10^{-38}\text{s}.$$

The following important remark is in order. By the end of inflation, the size of the universe has grown extraordinarily, which means that the pre-existing particles are now very diluted, and thus, the universe becomes extremely cold and has a very low entropy. As a consequence, in order to match it with the initial stage of the hot BB model, a *reheating mechanism is needed.* This mechanism is a very non-adiabatic process, in which an enormous amount of particles (practically the whole matter content of the universe, or primordial quark–gluon plasma) is created via quantum effects; after its thermalization, the universe also becomes reheated, and eventually matches with the starting conditions of the hot BB universe.

### 3.3. A Simple Way to Produce Inflation

We consider a universe filled out with a homogeneous scalar field, namely, $\phi(t)$, whose potential is $V(\phi)$. To start, we assume a static universe, and thus, the dynamical equation of a field can be readily obtained from the first law of thermodynamics:

$$d(\rho a^3) = -Pd(a^3). \tag{65}$$

Since we are assuming a static universe, $d(a^3) = 0$, and taking into account that the energy density is equal to the sum of its kinetic plus its potential part, i.e., $\rho = \frac{\dot{\phi}^2}{2} + V(\phi)$, the first law becomes:

$$d(\rho) = 0 \implies \ddot{\phi} + V_\phi(\phi) = 0, \tag{66}$$

which is the same equation as the one for a particle under the action of a potential that can be obtained from the Lagrangian:

$$\mathcal{L} = \left(\frac{\dot{\phi}^2}{2} - V(\phi)\right)a^3, \tag{67}$$

by using the Euler–Lagrange equations.

We now consider an expanding universe, where, from the Lagrangian (67), we obtain the dynamical equation:

$$\ddot{\phi} + 3H\dot{\phi} + V_\phi(\phi) = 0, \tag{68}$$

which could also be derived from the first law of thermodynamics (65):

$$(\ddot{\phi} + V_\phi(\phi))\dot{\phi}a^3 + 3H\left(\frac{\dot{\phi}^2}{2} + V(\phi)\right)a^3 = -3HPa^3. \tag{69}$$

Comparing this expression with the dynamical Equation (68), we conclude that the pressure when the universe is filled by a scalar field is:

$$P = \frac{\dot{\phi}^2}{2} - V(\phi). \tag{70}$$

Coming back to the dynamics, we have an autonomous second-order differential equation:

$$\ddot{\phi} + 3H\dot{\phi} + V_\phi(\phi) = 0, \tag{71}$$

where $H = \frac{1}{\sqrt{3}M_{pl}}\sqrt{\frac{\dot{\phi}^2}{2} + V(\phi)}$, for which, unfortunately, it is impossible to find analytic solutions, as in the case of a fluid. In fact, the system can only be solved numerically.

Recall, now, the equation for the acceleration $\frac{\ddot{a}}{a} = \dot{H} + H^2 = -\frac{H^2}{2}(1 + 3w_{eff})$, where $w_{eff} = P/\rho = -1 - \frac{2\dot{H}}{3H^2}$, and thus, the condition for an accelerated expansion is $\rho + 3P < 0$ or $w_{eff} < -1/3$. In a similar way, in terms of the Hubble rate and its derivative, we also see that the condition for acceleration is $-\frac{\dot{H}}{H^2} < 1$, and that acceleration ends when $-\frac{\dot{H}}{H^2} = 1$.

Therefore, a necessary condition to have accelerated expansion is $\dot{\phi}^2 < V(\phi)$, because:

$$\rho + 3P < 0 \Longrightarrow \dot{\phi}^2 < V(\phi). \tag{72}$$

Here, it is important to realize that the condition $P \cong -\rho$, and thus $\dot{H} \ll H^2$, is enough to have acceleration. Since the condition $P \cong -\rho$ is equivalent to $\dot{\phi}^2 \ll V(\phi)$, the successful realization of inflation requires keeping $\dot{\phi}^2$ small, as compared with $V(\phi)$ (the kinetic energy must be small compared with the potential one), during a sufficiently long time interval—more precisely, for at least 65 e-folds, but this will actually depend on the shape of the potential. In practice, one needs a very, very flat potential.

Then, with this condition, the FE becomes $H^2 \cong \frac{V(\phi)}{3M_{pl}^2}$. We also assume, during inflation, the condition $\ddot{\phi} \ll 3H\dot{\phi}$, and thus, the CE becomes $3H\dot{\phi} + V_\phi(\phi) \cong 0$. Note that, for a flat potential, one has $V_\phi(\phi) \cong 0 \Longrightarrow \dot{\phi} \cong 0$, and the resulting Hubble rate is nearly constant during this period, implying a nearly exponential growth of the volume of the universe during this epoch. This is exactly what we need to solve both the horizon and the flatness problems.

When both conditions,

$$\dot{\phi}^2 \ll V(\phi) \qquad \text{and} \qquad \ddot{\phi} \ll 3H\dot{\phi}, \tag{73}$$

are fulfilled, we say that the universe is in the *slow-roll regime*, which is mathematically an attractor (see, for instance, Section 3.7 of [6]), and the dynamical equations become:

$$H^2 \cong \frac{V(\phi)}{3M_{pl}^2} \qquad \text{and} \qquad 3H\dot{\phi} + V_\phi(\phi) \cong 0. \tag{74}$$

As an exercise, we now show that a necessary condition to have a slow-roll regime is that the *slow-roll parameters* satisfy:

$$\epsilon \equiv \frac{M_{pl}^2}{2}\left(\frac{V_\phi}{V}\right)^2 \ll 1 \quad \text{and} \quad \eta \equiv M_{pl}^2 \left|\frac{V_{\phi\phi}}{V}\right| \ll 1. \tag{75}$$

In fact, from (74), one has:

$$\frac{V_\phi}{3H} \cong -\dot{\phi} \Longrightarrow \frac{V_\phi^2}{9H^2} \cong \dot{\phi}^2 \ll V \Longrightarrow \frac{M_{pl}^2}{2}\frac{V_\phi^2}{V^2} \ll \frac{3}{2} \sim 1 \Longrightarrow \epsilon \ll 1, \tag{76}$$

where we have used, during the slow-roll regime, $H^2 \cong \frac{V}{3M_{pl}^2}$. Finally, to obtain the condition $\eta \ll 1$, we start by taking the temporal derivative of the slow-roll equation $3H\dot{\phi} + V_\phi(\phi) \cong 0$ to obtain:

$$\frac{V_{\phi\phi}}{V} \cong -\frac{3\dot{H}}{V} - \frac{3H\ddot{\phi}}{V\dot{\phi}} \Longrightarrow M_{pl}^2 \left| \frac{V_{\phi\phi}}{V} \right| \ll \frac{3|\dot{H}|M_{pl}^2}{V} + 3, \tag{77}$$

where we have used the slow-roll condition $\ddot{\phi} \ll 3H\dot{\phi}$ and the Friedmann equation in the slow-roll regime. In addition, from the Raychaudhuri equation, $\dot{H} = -\frac{\dot{\phi}^2}{2M_{pl}^2}$, we obtain:

$$M_{pl}^2 \left| \frac{V_{\phi\phi}}{V} \right| \ll \frac{3\dot{\phi}^2}{2V} + 3 \ll \frac{9}{2} \sim 1 \Longrightarrow \eta \ll 1, \tag{78}$$

where we have it that, during the slow-roll regime, the kinetic energy is smaller than the potential one, i.e., $\dot{\phi}^2 \ll V$.

At this point, we want to understand a bit better the meaning of the slow-roll conditions. The first condition, $\dot{\phi}^2 \ll V(\phi)$, is clear enough for producing acceleration because it implies $w_{eff} \cong -1$. To understand the other one, $\ddot{\phi} \ll 3H\dot{\phi}$, we consider a harmonic oscillator with friction:

$$\ddot{x} + \gamma\dot{x} + \omega^2 x = 0, \qquad \text{with} \qquad \gamma > 0, \tag{79}$$

which corresponds to the movement of a particle under the influence of the potential $V(x) = \omega^2 x^2/2$.

The general solution is given by:

$$x(t) = C_+ e^{\lambda_+ t} + C_- e^{\lambda_- t}, \tag{80}$$

where

$$\lambda_\pm = \frac{-\gamma \pm \sqrt{\gamma^2 - 4\omega^2}}{2}. \tag{81}$$

Observe that the friction term $\gamma\dot{x}$ damps the velocity of the particle. In addition, for a very flat potential, i.e., $\omega^2 \ll \gamma^2$, one obtains:

$$\lambda_+ \cong -\omega^2/\gamma \qquad \text{and} \qquad \lambda_- \cong -\gamma. \tag{82}$$

So, at late time, the solution becomes $x(t) \cong C_+ e^{-\omega^2 t/\gamma}$, which means that the solution of the following equation (the slow-roll solution):

$$\gamma\dot{x} + \omega^2 x = 0, \tag{83}$$

which is given by $e^{-\omega^2 t/\gamma}$, is an attractor.

Having now better grasped the slow-roll conditions, we introduce another pair of slow-roll parameters, namely:

$$\bar{\epsilon} \equiv -\frac{\dot{H}}{H^2}, \qquad \bar{\eta} \equiv 2\bar{\epsilon} - \frac{\dot{\bar{\epsilon}}}{2H\bar{\epsilon}}. \tag{84}$$

It is not difficult to see that, using the Friedmann and the Raychaudhuri equations, the condition $\bar{\epsilon} \ll 1$ implies $\dot{\phi}^2 \ll V(\phi)$ (first slow-roll condition). In fact,

$$-\dot{H} \ll H^2 \Longrightarrow \frac{\dot{\phi}^2}{2} \ll \frac{\dot{\phi}^2 + 2V}{6} \Longrightarrow \dot{\phi}^2 \ll V(\phi), \tag{85}$$

where we have used the Friedmann and the Raychaudhuri equations.

On the other hand, the condition $\bar{\eta} \ll 1$, together with $\bar{\epsilon} \ll 1$, implies $\dot{\bar{\epsilon}} \ll 2H\bar{\epsilon}$. Next, we make use of:

$$\dot{\bar{\epsilon}} = -\frac{\ddot{H}}{H^2} + 2H\bar{\epsilon}^2, \tag{86}$$

in order to obtain:

$$-\frac{\ddot{H}}{H^2} \ll 2H(\bar{\epsilon} - \bar{\epsilon}^2) \sim 2H\bar{\epsilon}, \qquad \text{because} \qquad \bar{\epsilon}^2 \ll \bar{\epsilon}. \tag{87}$$

Using the definition of $\bar{\epsilon}$, the last condition is equivalent to:

$$-\ddot{H} \ll -2H\dot{H}, \tag{88}$$

and taking into account the RE, $\dot{H} = -\frac{\dot{\phi}^2}{2M_{pl}^2} \Longrightarrow \ddot{H} = -\frac{\ddot{\phi}\dot{\phi}}{M_{pl}^2}$, we obtain:

$$\ddot{\phi} \ll H\dot{\phi} \sim 3H\dot{\phi} \qquad \text{(second slow-roll condition).} \tag{89}$$

In this way, we have proved that sufficient conditions to have the slow-roll regime are $\bar{\epsilon} \ll 1$ and $\bar{\eta} \ll 1$.

Finally we show that during the slow-roll regime one has $\bar{\epsilon} \cong \epsilon$ and $\bar{\eta} \cong \eta$. Effectively, during slow-roll, we have:

$$\bar{\epsilon} \cong \frac{3\dot{\phi}^2}{2V} \cong \frac{V_\phi^2}{6H^2V} \cong \epsilon. \tag{90}$$

On the other hand, a simple calculation leads to:

$$\bar{\eta} = \bar{\epsilon} - \frac{\ddot{H}}{2H\dot{H}} \cong \epsilon - \frac{\ddot{H}}{2H\dot{H}}. \tag{91}$$

Next, we have:

$$\frac{\ddot{H}}{2H\dot{H}} \cong -\frac{1}{2H\dot{H}}\left(-\frac{V_{\phi\phi}\dot{\phi}^2}{3HM_{pl}^2} + \frac{V_\phi \dot{H}\dot{\phi}}{3H^2 M_{pl}^2}\right) \cong -\frac{V_{\phi\phi}}{3H} - \frac{V_\phi \dot{\phi}}{6H^2 M_{pl}^2} \cong -\eta + \epsilon, \tag{92}$$

where we have used both slow-roll equations.

Thus, in practice, the slow-roll regime is equivalent to $\epsilon \ll 1$ and $\eta \ll 1$.

### 3.4. On the Number of E-Folds

We will now obtain the minimum number of necessary e-folds during the slow-roll regime. Since $H = \frac{d\ln a}{dt}$, from the CE, we have:

$$H = \frac{d\ln a}{dt} = \dot{\phi}\frac{d\ln a}{d\phi} = -\frac{V_\phi}{3H}\frac{d\ln a}{d\phi} \Longleftrightarrow 3H^2 = -V_\phi \frac{d\ln a}{d\phi},$$

and now, using the FE, one obtains:

$$\frac{d\ln a}{d\phi} = \frac{1}{\sqrt{2\epsilon}M_{pl}}, \tag{93}$$

whose solution is:

$$a_e = a_b \exp\left(\frac{1}{M_{pl}}\left|\int_{\phi_b}^{\phi_e}\frac{d\phi}{\sqrt{2\epsilon}}\right|\right) \Longrightarrow N = \frac{1}{M_{pl}}\left|\int_{\phi_b}^{\phi_e}\frac{d\phi}{\sqrt{2\epsilon}}\right|. \tag{94}$$

In addition, since in the slow-roll regime one has $\epsilon \cong -\frac{\dot{H}}{H^2}$, from the acceleration equation $\frac{\ddot{a}}{a} = \dot{H} + H^2$, one can conclude that inflation ends when $\epsilon = 1$.

As an example, we consider the quadratic potential $V(\phi) = \frac{1}{2}m^2\phi^2$, where a simple calculation leads to:

$$\epsilon = \eta = \frac{2M_{pl}^2}{\phi^2}; \tag{95}$$

therefore, the slow-roll regime is satisfied when $\phi > \sqrt{2}M_{pl}$, and inflation ends at $\phi_e = \sqrt{2}M_{pl}$.

On the other hand,

$$N = \frac{1}{M_{pl}} \int_{\phi_e}^{\phi_b} \frac{\phi}{2M_{pl}} d\phi = \frac{1}{4M_{pl}^2}(\phi_b^2 - 2M_{pl}^2),$$

which means that, to obtain more than 65 e-folds, the beginning of inflation has to occur at $\phi_b > \sqrt{262}M_{pl}$.

To finish this section, it is also important to calculate the last number of e-folds, that is, the number of them from the horizon-crossing time to the end of inflation.

The horizon crossing refers to the moment when the "pivot scale", namely, $k_*$ in comoving coordinates, leaves the Hubble horizon; that is, when $k_* = a_* H_* \iff \frac{1}{H_*} = \frac{a_*}{k_*}$, where the value of the Hubble rate at this moment is called the *scale of inflation*.

The physical value of the "pivot scale" at present time is usually chosen as (see, for instance, [23]) $k_{phys}(t_0) = \frac{k_*}{a_0} = 0.05 \text{ Mpc}^{-1} \cong 10^{-58} M_{pl} \sim 10^2 H_0$, where we use the value $H_0 \cong 6 \times 10^{-61} M_{pl}$.

To calculate the e-folds from the horizon-crossing moment to the end of inflation, we start with the relation $a_{END} = e^{N_*} a_* = e^{N_*} \frac{k_*}{H_*}$, which can be written as follows:

$$\frac{k_*}{a_0 H_0} = e^{-N_*} \frac{H_* a_{END}}{a_0 H_0} \implies \frac{k_*}{a_0 H_0} = e^{-N_*} \frac{H_*}{H_0} \frac{a_{END}}{a_{rad}} \frac{a_{rad}}{a_{matt}} \frac{a_{matt}}{a_0}, \tag{96}$$

where *rad* (resp. *matt*) denotes the beginning of radiation (resp. of matter domination, i.e, at the matter–radiation equality).

To simplify, we will assume that, from the end of inflation to the beginning of kination, the EoS parameter $w$ is constant. Then, we have:

$$\left(\frac{a_{END}}{a_{rad}}\right)^{3(1+w)} = \frac{\rho_{rad}}{\rho_{END}}, \qquad \left(\frac{a_{rad}}{a_{matt}}\right)^4 = \frac{\rho_{matt}}{\rho_{rad}}. \tag{97}$$

Therefore:

$$N_* = -5.52 + \ln\left(\frac{H_*}{H_0}\right) + \frac{1}{4}\ln\left(\frac{\rho_{matt}}{\rho_{rad}}\right) + \frac{1}{3(1+w)}\ln\left(\frac{\rho_{rad}}{\rho_{END}}\right) + \ln\left(\frac{a_{matt}}{a_0}\right). \tag{98}$$

To perform the calculations, we consider a power-law potential of the form $V(\phi) = V_0\left(\frac{\phi}{M_{pl}}\right)^{2n}$. Using the virial theorem, one can show that, after the end of inflation, when the inflaton oscillates, the effective EoS parameter is given by $w = \frac{n-1}{n+1}$.

Now, we need the spectral index of scalar perturbations, defined by $n_s = 1 - 6\epsilon_* + 2\eta_*$, and also the ratio of tensor to scalar perturbations $r = 16\epsilon_*$, where, once again, the star denotes that the quantities are evaluated at horizon crossing. As a simple exercise, it can be shown that, for our power-law potential, the slow-roll parameter $\epsilon_*$ and the spectral index are related by:

$$\epsilon_* = \frac{n(1 - n_s)}{2(n+1)}. \tag{99}$$

From the slow-roll parameter $\epsilon$, one can calculate the value of the energy density at the end of inflation; imposing that inflation ends when $\epsilon = 1$, we obtain:

$$\epsilon = 2n^2 \left( \frac{M_{pl}}{\phi} \right)^2 \implies \phi_{END} = \sqrt{2}nM_{pl} \implies V(\phi_{END}) = 2^n n^{2n} V_0, \tag{100}$$

taking into account that inflation ends when:

$$w_{eff} = -\frac{1}{3} \implies \dot{\phi}^2_{END} = V(\phi_{END}) \implies \rho_{END} = \frac{3}{2}V(\phi_{END}) = 3 \times 2^{n-1} n^{2n} V_0. \tag{101}$$

We also need the power spectrum of scalar perturbations, defined as:

$$\mathcal{P}_\zeta = \frac{H_*^2}{8\pi^2 M_{pl}^2 \epsilon_*} \sim 2 \times 10^{-9} \iff H_*^2 = 16\pi^2 \times 10^{-9} \epsilon_* M_{pl}^2, \tag{102}$$

which is essential to calculate the value of $V_0$. Indeed, the square of the scale of inflation is given by:

$$H_*^2 = \frac{V(\phi_*)}{3M_{pl}^2} = \frac{V_0}{3M_{pl}^2} \left( \frac{2}{\epsilon_*} \right)^n n^{2n}, \tag{103}$$

where we have used that, at horizon crossing, $\phi_* = \sqrt{\frac{2}{\epsilon_*}} n M_{pl}$. Inserting the value of the square of the scale of inflation into the formula of the power spectrum, we finally obtain:

$$V_0 = \frac{96\pi^2}{n^{2n}} \times 10^{-9} \left( \frac{\epsilon_*}{2} \right)^{n+1} M_{pl}^4 \implies \rho_{END} = 72\pi^2 \times 10^{-9} \epsilon_*^{n+1} M_{pl}^4. \tag{104}$$

Next, we use the Stefan–Boltzmann law at the beginning of radiation and at the matter–radiation equality:

$$\rho_{rad} = \frac{\pi^2}{30} g_{rad} T_{rh}^4, \qquad \rho_{matt} = \frac{\pi^2}{15} g_{matt} T_{matt}^4, \tag{105}$$

where $g_{rad} = 106.75$ is the number of effective degrees of freedom for the standard model, $g_{matt} = 3.36$ is the number of the effective degrees of freedom at matter–radiation equality, $T_{rh}$ is the reheating temperature (the temperature of the universe at the beginning of the radiation era), and $T_{matt}$ is the temperature of the universe at matter–radiation equality. (Note that, at matter–radiation equality, the energy density of radiation is the same as that for matter; for this reason, the factor $1/15$ appears in the energy density at that moment).

Finally, we need use the adiabatic evolution of the universe after the matter–radiation equality, $a_0 T_0 = a_{matt} T_{matt}$. Inserting all these quantities in (98), one obtains:

$$N_* \cong -15 - \frac{5}{1+w} + \left( \frac{1}{2} - \frac{n+1}{3(1+w)} \right) \ln \epsilon_* + \ln \left( \frac{T_0 M_{pl}}{T_{rh} H_0} \right) + \frac{4}{3(1+w)} \ln \left( \frac{T_{rh}}{M_{pl}} \right), \tag{106}$$

and using $H_0 \sim 6 \times 10^{-61} M_{pl}$ and $T_0 \cong 2.73\,\text{K} \sim 10^{-31} M_{pl}$, the number of e-folds finally becomes:

$$N_* \cong 52 - \frac{5}{1+w} + \left( \frac{1}{2} - \frac{n+1}{3(1+w)} \right) \ln \epsilon_* + \left( 1 - \frac{4}{3(1+w)} \right) \ln \left( \frac{M_{pl}}{T_{rh}} \right). \tag{107}$$

To finish, inserting the value of $w = \frac{n-1}{n+1}$, $\epsilon_* = \frac{n(1-n_s)}{2(n+1)}$, and taking the central value of the spectral index $n_s \cong 0.96$, one obtains the last number of e-folds as a function of $n$:

$$N_*(n) \cong 52 + \frac{n^2 - 6n - 4}{2n} - \frac{n^2 - n + 1}{6n} \ln\left(\frac{n}{2(n+1)}\right) + \frac{n-2}{3n} \ln\left(\frac{M_{pl}}{T_{rh}}\right). \qquad (108)$$

So, for a quadratic potential, the number of e-folds as a function of the reheating temperature is $N_*(1) \cong 48 - \frac{2}{3} \ln\left(\frac{M_{pl}}{T_{rh}}\right)$ and, for a quartic potential, one has $N_*(2) \cong 49$, which does not depend on the reheating temperature, because for a quartic potential, when the inflaton starts to oscillate, $w = 1/3$, that is, the universe enters the radiation phase.

### 3.5. Different Reheating Mechanisms

1. In the case that the potential has a deep well, at the end of inflation, the inflaton field starts to oscillate in this deep well, and releases its energy by creating particles [43,44,76]. This happens in standard inflation, but after the discovery of the current cosmic acceleration, other models containing monotonic potentials appeared, and thus, since the inflaton field cannot oscillate in this case, other mechanisms to reheat the universe were proposed.

2. When the potential is a monotonous function, particles could be created via the so-called *instant preheating* developed by Felder, Kofman, and Linde [46]. In that case, a quantum scalar field with a very light bare mass is coupled with the inflaton, the adiabatic regime breaks after the end of inflation, and particles with an effective very heavy mass are copiously created. The energy density of these particles could never dominate the one of the background, because in that case, another undesirable inflationary period would appear; this is the reason why these particles have to decay in lighter ones well before they can dominate. Once the decay is finished, the universe becomes reheated, at a temperature close to $10^9$ GeV, thus matching with the hot BB model.

3. For a monotonous potential, containing an abrupt phase transition from the end of inflation to a regime where all the energy density is kinetic (named *kination phase* [29] or *deflationary phase* in [39]), superheavy particles [40,41,77,78] and also lighter ones [30–38,42,79,80] can be created via *gravitational particle production*. The problem of reheating via the production of light particles is that undesirable polarization effects could appear, which would disturb the evolution of the inflaton field during the slow-roll period (see, for a detailed explanation, [46]). On the contrary, these polarization effects, during inflation, can be neglected when one considers the production of super-heavy particles, which have to decay in lighter ones, to obtain, after thermalization, a hot radiation-dominated universe.

4. The *curvaton* mechanism. In addition to the inflaton, there is another massive field, named the *curvaton*, which becomes sterile during inflation. At the end of this period, this curvaton field, whose potential has a deep well, starts to oscillate, decaying into lighter particles [81–85].

To finish this section, we should add that the reheating parameters (especially the reheating equation of state parameter) are not sufficiently well constrained. Among other contributions, some possible ways to constrain the reheating EoS parameter have recently been proposed, involving magnetogenesis or primordial gravitational waves. The reheating era has been argued to have a considerable effect on the primordial magnetic field as well as on primordial GWs, which in turn help to extract some viable constraints on the reheating parameters. This has been carried out for two different reheating mechanisms. In the case of GWs, it has also been shown that a late reheating phase helps to improve the fit of the NANOGrav observational data [86–89].

## 4. The Current Cosmic Acceleration

*4.1. The Cosmological Constant*

The cosmological constant, $\Lambda$, was introduced by Albert Einstein (1917) in order to obtain a static model for our universe (at that time, because of very reasonable physical considerations, everybody believed that the universe was static; see all details in, e.g., [1–4]). The introduction of $\Lambda$ modifies the FE as follows:

$$H^2 = \frac{\rho}{3M_{pl}^2} - \frac{\kappa}{a^2} + \frac{\Lambda}{3}. \tag{109}$$

From this equation, we see at once that the introduction of $\Lambda$ is equivalent to the addition fo a new component to the energy content of the universe, with an energy density $\rho_\Lambda = \Lambda M_{pl}^2$, and, from the CE, with pressure $P_\Lambda = -\rho_\Lambda$. Thus, a cosmological constant with a positive sign acts against gravitation: ($w_\Lambda = P_\Lambda / \rho_\Lambda = -1$).

Einstein also considered the RE:

$$\dot{H} = -\frac{\rho + P}{2M_{pl}^2} + \frac{\kappa}{a^2}. \tag{110}$$

Then, for a matter-dominated universe, $P = 0$, a static solution, $H = \dot{H} = 0$, as the one Einstein was looking for, must satisfy:

$$\frac{\rho}{3M_{pl}^2} - \frac{\kappa}{a^2} + \frac{\Lambda}{3} = 0 \quad \text{and} \quad \frac{\rho}{2M_{pl}^2} = \frac{\kappa}{a^2}. \tag{111}$$

From these equations, we see that a static universe has to be closed, i.e., $\kappa = 1$. In Einstein's static model, the energy density and the radius of the universe are given by:

$$\rho = 2\Lambda M_{pl}^2 \quad \text{and} \quad a = \Lambda^{-1/2}, \tag{112}$$

respectively.

Unfortunately, from the acceleration equation $\frac{\ddot{a}}{a} = -\frac{\rho}{6M_{pl}^2} + \frac{\Lambda}{3}$, one can show that Einstein's static model is unstable; that is, with a simple sneeze, his universe collapses. This was the real problem of that model, not actually the fact that it did not describe the expansion of the universe. The case is that Einstein did not realize this problem, which was later noted by Lemaître and by Eddington, among others.

To further show the instability, we combine the acceleration and Friedmann equations to obtain:

$$\ddot{a} = -\frac{\dot{a}^2}{2a} - \frac{1}{2a} + \frac{\Lambda a}{2}, \tag{113}$$

which, as a dynamical system, can be written as:

$$\begin{cases} \dot{a} = b \\ \dot{b} = -\frac{b^2}{2a} - \frac{1}{2a} + \frac{\Lambda a}{2}, \end{cases} \tag{114}$$

where $b = \dot{a}$. It is clear that the Einstein solution corresponds to the fixed point $a = \Lambda^{-1/2}$ and $b = 0$.

To study the stability of this fixed point, one can linearize the system around it, thus obtaining the matrix:

$$\begin{pmatrix} 0 & 1 \\ \Lambda & 0 \end{pmatrix}, \tag{115}$$

which has a negative determinant equal to $-\Lambda$; this means that the fixed point is a saddle point, and is, thus, unstable.

Presently, however, the introduction of the CC by Einstein is no longer seen as a horrible mistake but, quite the contrary, to have had extremely positive consequences. Namely, now that we know that the universe expansion accelerates, the CC could be a most natural candidate for dark energy, explaining the current cosmic acceleration within the standard cosmological model without any extra addition. To this end, we consider a flat universe filled by matter and with the CC. Since $\rho$ scales as $a^{-3}$ and $\rho_\Lambda$ is constant, the constituent equations have the fixed point $\rho = 0$, $H = \sqrt{\frac{\Lambda}{3}}$. This is de Sitter's solution, which naturally appears at late times, and since $\dot{H} = 0$, i.e., $w_{eff} = -1$, this means that it truly depicts an accelerating universe.

At present, 70% of the energy density of the universe is dark and the ordinary matter/energy amount only represents some 30% of the total. Now, by using the CC, we have $H_0^2 \cong \frac{\Lambda}{3}$. Since $H_0 \sim 6 \times 10^{-61} M_{pl}$, we obtain a very small value for the CC, namely, $\Lambda \sim 10^{-120} M_{pl}^2$. Involving quantum considerations, this number appears to be extremely small when we compare it with the expected contributions to the CC coming from the unavoidable quantum vacuum fluctuations of the different fields present in the universe. In order to describe our present universe, using the CC as a source of dark energy, we have to fine-tune $\Lambda$ extremely well, down to some hundreds of orders of magnitude (what has been sometimes called the highest discrepancy between theory and observations ever encountered in physics).

*4.2. Quintessential Inflation*

The main idea in quintessential inflation goes as follows. The inflaton field could actually be responsible not only for the very early, but also for the late-time acceleration of the universe. To obtain a successful reheating stage, an abrupt phase transition must occur from the end of inflation to the beginning of kination (the epoch when the energy density of the field was (almost) exclusively kinetic, i.e., $w_{eff} = 1$). There, the adiabatic evolution is broken in order to create enough superheavy particles, whose energy density ($\langle \rho \rangle \sim a^{-3}$) will eventually dominate the one of the inflaton field ($\rho_\varphi \sim a^{-6}$) after decaying into lighter particles, in order to match with the hot BB conditions and conveniently enter into the radiation phase in a smooth way (the kination phase ends or the radiation era starts when $\rho_\varphi \sim \langle \rho \rangle$). Then, the universe slowly cools down and particles become non-relativistic, thus entering in the matter domination era.

Finally, "close" to present time, the remaining energy density of the inflaton field starts to dominate once again, as a new form of dark energy, termed *quintessence*, which is able to reproduce the current cosmic acceleration, dominates again the energy balance, but now in a much more equilibrated way. The questions: "why this is so?" and "why does this happen precisely now?" are very important ones, and the present standard cosmological model has been unable to answer them up to now.

**Remark 4.** *The unification of the early and late-time acceleration of our universe could also be obtained in other theories, such as modified gravity [90], in $F(R)$ gravity [91], or in $F(R, T)$ gravity [92], where, here, $T$ denotes the trace of the stress tensor.*

An important observation is also that, owing to the kination regime, the number of "last" e-folds is larger than in the case of standard inflation: in most of the models, it ranges, more or less, between 60 and 70. Effectively, in [93], for a model of QI, the number of e-folds is given by:

$$N + \ln N \cong 55 - \frac{1}{3} \ln \left( \frac{T_{reh}}{M_{pl}} \right), \tag{116}$$

and, taking into account that the scale of nucleosynthesis is 1 MeV [94] and in order to avoid the late-time decay of gravitational relic products such as moduli fields or gravitinos, which could jeopardize the success of nucleosynthesis, one needs temperatures lower than $10^9$ GeV [95,96]. So, we will assume that 1 MeV $\leq T_{reh} \leq 10^9$ GeV, which leads to constraining the number of e-folds to $58 \lesssim N \lesssim 67$.

The first model of quintessential inflation (QI) was introduced by Peebles and Vilenkin in their seminal paper entitled "Quintessential Inflation" [28] (see [97] for a review and [98–107] for other quintessential inflation models, such as exponential models, as in [102], where, to match with the current observational data, the authors assume that the inflaton field is non-minimally coupled with massive neutrinos), and it is defined by the potential:

$$
V(\varphi) = \begin{cases} \lambda \left( \varphi^4 + M^4 \right) & \text{for} \quad \varphi \leq 0 \\ \lambda \dfrac{M^8}{\varphi^4 + M^4} & \text{for} \quad \varphi \geq 0. \end{cases} \tag{117}
$$

Here, $\lambda$ is a dimensionless parameter and $M \ll M_{pl}$ is a very small mass, as compared with the reduced Planck mass. An abrupt phase transition takes place at $\varphi = 0$, where the fourth derivative of $V$ is discontinuous.

The first part of the potential, the quartic potential, is the one responsible for inflation, while the quintessence tail, the inverse power-law potential, is responsible for the current cosmic acceleration.

As we will see, $\lambda \cong 9 \times 10^{-11}$ is obtained from the power spectrum of scalar perturbations and $M \sim 200$ TeV has to be calculated numerically, using the observational data $\Omega_{\varphi,0} \equiv \frac{\rho_{\varphi,0}}{3 H_0^2 M_{pl}^2} \cong 0.7$.

It is, here, important to recall that the following quantities, which we have already defined, can be actually measured:

1. The power spectrum of scalar perturbations:

$$
P_\zeta = \frac{H_*^2}{8 \pi^2 M_{pl}^2 \epsilon_*} \sim 2 \times 10^{-9}, \tag{118}
$$

where the star means that the quantities are evaluated at the horizon crossing;
2. The spectral index, $n_s \cong 1 - 6\epsilon_* + 2\eta_*$. Its central value is $n_s \cong 0.9649$;
3. The ratio of tensor to scalar perturbations, $r = 16\epsilon_*$. Observational data lead to the constraint $r \leq 0.1$.

As an example, for a quartic potential $V(\phi) = \lambda \varphi^4$, the inflationary piece of the PV model, leads to the relation:

$$
n_s = 1 - \frac{3}{16} r. \tag{119}
$$

However, recent observational data yield $n_s = 0.9649 \pm 0.0042$, at $1\sigma$ confidence level, and $r \leq 0.1$. This means that the Peebles–Vilenkin model is not compatible with the current observational data at $2\sigma$ C.L., because from the model one obtains the bound: $r \geq 0.1424$.

### 4.3. Improved Versions of QI

An improved version of QI is the Higgs inflation + inverse power-law, which is given by the potential:

$$
V(\varphi) = \begin{cases} \lambda M_{pl}^4 \left( 1 - e^{\sqrt{\frac{2}{3}} \frac{\varphi}{M_{pl}}} \right)^2 + \lambda M^4 & \text{for} \quad \varphi \leq 0 \\ \lambda \dfrac{M^8}{\varphi^4 + M^4} & \text{for} \quad \varphi \geq 0. \end{cases} \tag{120}
$$

For this potential, it is easy to show that one obtains the relation:

$$r = 3(1 - n_s)^2, \tag{121}$$

which perfectly fits in the joint contour at $2\sigma$ C.L., because in that case, we have $r \leq 0.006$.

Now, we calculate the values of the two parameters of the model. Note first that the scale of inflation, i.e., the value of the Hubble parameter at the horizon crossing, is $H_*^2 \cong \frac{\lambda}{3} M_{pl}^2$. Then, with the power spectrum for scalar perturbations $\mathcal{P}_\zeta = \frac{H_*^2}{8\pi^2 M_{pl}^2 \epsilon_*} \sim 2 \times 10^{-9}$, the formula $r = 3(1 - n_s)^2$, the relation $r = 16\epsilon_*$, and the central value of the spectral index, $n_s \cong 0.96$, one can show that $\lambda \cong 9 \times 10^{-11}$.

Finally, from the observational data $\Omega_{\varphi,0} = \frac{V(\varphi_0)}{3H_0^2 M_{pl}^2} \cong 0.7$, with $\varphi_0 \cong 30 M_{pl}$ (we will see later that this is approximately the value of the scalar field at present time), one obtains that $M \sim 10^5$ GeV.

It is important to remark that both the Peebles–Vilenkin model and also this improved version should be viewed simply as phenomenological models, as first steps to understand QI. More physically grounded models are the following.

### 4.3.1. Lorentzian Quintessential Inflation

Based on the Lorentzian (or Cauchy, for mathematicians) distribution, one considers the following ansatz:

$$\epsilon(N) = \frac{\bar{\xi}}{\pi} \frac{\Gamma/2}{N^2 + \Gamma^2/4}, \tag{122}$$

where $\epsilon$ is the slow-roll parameter, $N$ is the number of e-folds, and $\xi$ and $\Gamma$ are the amplitude and width of the Lorentzian distribution, respectively.

From the previous ansatz, one obtains the potential: [108–110]

$$V(\varphi) = \lambda M_{pl}^4 \exp\left[-\frac{2\gamma}{\pi} \arctan\left(\sinh\left(\gamma\varphi/M_{pl}\right)\right)\right], \tag{123}$$

where $\lambda$ is a dimensionless parameter and $\gamma$ is defined by

$$\gamma \equiv \sqrt{\frac{\pi}{\Gamma\xi}}.$$

The model depends on these two parameters, and in order to match it with current observational data, one has to impose them to take the values $\lambda \sim 10^{-69}$ and $\gamma \cong 122$. This leads, then, to a successful inflation model that, at late times, yields an eternal acceleration with effective EoS parameter equal to $-1$. It is, thus, indistinguishable from the simple CC in this regime.

### 4.3.2. $\alpha$-Attractors in Quintessential Inflation

In that case, the corresponding potential, combined with a standard exponential potential, is obtained from a Lagrangian motivated by super-gravity and corresponding to a non-trivial Kähler manifold.

The Lagrangian provided by super-gravity theories is [111]:

$$\mathcal{L} = \frac{1}{2} \frac{\dot{\phi}^2}{(1 - \frac{\phi^2}{6\alpha})^2} M_{pl}^2 - \lambda M_{pl}^4 e^{-\kappa\phi}, \tag{124}$$

where $\phi$ is a dimensionless scalar field, and $\kappa$ and $\lambda$ are positive dimensionless constants. In order that the kinetic term acquires the canonical form, i.e., $\frac{1}{2}\dot{\varphi}^2$, one can redefine the scalar field as follows:

$$\phi = \sqrt{6\alpha}\,\tanh\left(\frac{\varphi}{\sqrt{6\alpha}\,M_{pl}}\right),\tag{125}$$

thus obtaining the following potential [112,113]:

$$V(\varphi) = \lambda M_{pl}^4 e^{-n\tanh\left(\frac{\varphi}{\sqrt{6\alpha}M_{pl}}\right)},\tag{126}$$

which is depicted in Figure 1, where we have introduced the notation $n \equiv \kappa\sqrt{6\alpha}$; by taking $\alpha \sim 10^{-2}$, chosen to match with observational data, we obtain the current cosmic acceleration when $n \sim 10^2$ and $\lambda \sim 10^{-66}$.

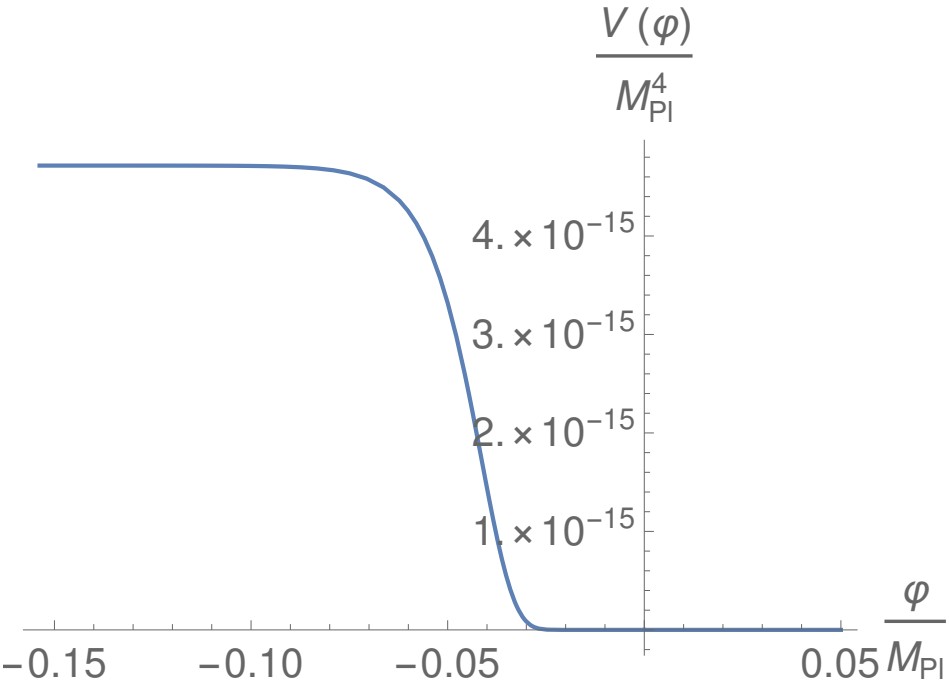

**Figure 1.** The $\alpha$-attractor potential. In Lorentzian QI, the corresponding potential has a similar shape.

Finally, for $\alpha$-attractors, the spectral index and the tensor/scalar ratio, as a function of the number of e-folds, have the following simple form:

$$n_s \cong 1 - \frac{2}{N}, \qquad r \cong \frac{12\alpha}{N^2},\tag{127}$$

which, for the usual number of e-folds in quintessential inflation ($65 \le N \le 75$), matches correctly with the observational data $n_s = 0.9649 \pm 0.0042$ and $r \le 0.1$ at 1-$\sigma$ confidence level.

In fact, for $N = 65$, the value of the ratio of tensor to scalar perturbations is very small, around $r \cong 10^{-3}\alpha$, and smaller for $N > 65$.

### 4.4. Evolution of the Dynamical System

To understand the evolution of the universe in QI, we deal with the model (120). First, we calculate the energy density at the end of inflation. Since inflation ends when $\epsilon \cong -\frac{\dot{H}}{H^2} = 1 \Longrightarrow w_{eff} = -1/3$, one has $\rho_{END} = \frac{3}{2}V(\varphi_{END})$.

On the other hand, since $\epsilon = \frac{M_{pl}^2}{2}\left(\frac{V_\varphi}{V}\right)^2$, at the end of inflation, one has $\varphi_{END} = \frac{1}{2}\sqrt{\frac{3}{2}}\ln(3/4)M_{pl}$.

Note that the kination regime starts, approximately, for $\varphi_{kin} = 0 > \varphi_{END}$, when the potential is negligible. Assuming, as usual, that there is no energy drop between the end of inflation and the beginning of kination, we obtain $\dot\varphi_{kin} = \sqrt{3V(\varphi_{END})}$. Then, the initial conditions at the beginning of kination are:

$$\left(\varphi_{kin} = 0, \dot\varphi_{kin} = \sqrt{\frac{3\lambda}{4}}(2-\sqrt3)M_{pl}^2\right) \Longrightarrow H_{kin} = \frac{\sqrt\lambda}{2\sqrt2}(2-\sqrt3)M_{pl}, \tag{128}$$

and thus, $H_{kin} \cong 10^{-6}M_{pl}$.

During kination, the effective EoS parameter is given by $w_{eff} = 1$, because all energy density is kinetic. So, during this period, the Hubble rate evolves as $H(t) = \frac{1}{3t}$. Then, since the value of the potential is very small as compared with the kinetic energy, one can safely disregard it in the FE, thus obtaining:

$$\frac{\dot\varphi^2}{2} = \frac{M_{pl}^2}{3t^2} \Longrightarrow \dot\varphi(t) = \sqrt{\frac{2}{3}}\frac{M_{pl}}{t} \Longrightarrow \varphi(t) = \sqrt{\frac{2}{3}}M_{pl}\ln\left(\frac{t}{t_{kin}}\right). \tag{129}$$

Therefore, we end up with:

$$\left(\varphi(t) = \sqrt{\frac{2}{3}}M_{pl}\ln\left(\frac{H_{kin}}{H(t)}\right), \dot\varphi(t) = \sqrt6 H(t)M_{pl}\right). \tag{130}$$

Now, for simplicity, we shall assume that the created particles during the phase transition from the end of inflation to the beginning of kination decay in lighter ones before the end of kination occurs, and that they thermalize almost instantaneously. Under these circumstances, the universe becomes reheated at the end of the kination phase (the energy density of the inflaton is the same as the one of the created particles), just when the energy density of the created particles starts to dominate. As a consequence, at reheating time, we have:

$$\left(\varphi_{rh} = \sqrt{\frac{2}{3}}M_{pl}\ln\left(\frac{H_{kin}}{H_{rh}}\right), \dot\varphi_{rh} = \sqrt6 H_{rh}M_{pl}\right). \tag{131}$$

Using that $H_{rh}^2 \cong \frac{2\rho_{rh}}{3M_{pl}^2}$, where $\rho_{rh} = \frac{\pi^2}{30}g_* T_{rh}^4$ is the energy density of radiation (Stefan–Boltzmann's law), one can write the value of the inflaton field and its derivative as a function of the reheating temperature.

Next, during radiation domination $w_{eff} = 1/3$, we have $H(t) = \frac{1}{2t}$. Taking into account that, in the radiation epoch, the potential energy may be disregarded too (since it is negligible as compared with the kinetic one), the CE becomes:

$$\ddot\varphi + \frac{1}{2t}\dot\varphi = 0. \tag{132}$$

Integrating, now, this equation, one can see that, during the radiation-dominated epoch, one has:

$$\varphi(t) = \varphi_{rh} + 2\dot\varphi_{rh}t_{rh}\left(1 - \sqrt{\frac{t_{rh}}{t}}\right) = \varphi_{rh} + \frac{\dot\varphi_{rh}}{H_{rh}}\left(1 - \sqrt{\frac{H(t)}{H_{rh}}}\right) \quad \text{and} \quad \dot\varphi(t) = \dot\varphi_{rh}\left(\frac{H(t)}{H_{rh}}\right)^{3/2}.$$

Therefore, at the matter–radiation equality, we will have the following initial conditions:

$$\left( \varphi_{eq} = \varphi_{rh} + \frac{\dot{\varphi}_{rh}}{H_{rh}} \left( 1 - \sqrt{\frac{H_{eq}}{H_{rh}}} \right), \dot{\varphi}_{eq} = \dot{\varphi}_{rh} \left( \frac{H_{eq}}{H_{rh}} \right)^{3/2} \right).$$

At matter–radiation equality, we have $H_{eq}^2 \cong \frac{2\rho_{eq}}{3M_{pl}^2}$, where $\rho_{eq} = \frac{\pi^2}{30} g_{eq} T_{eq}^4$, and the number of degrees of freedom is $g_{eq} = 3.36$.

Now, it is important to know the corresponding initial conditions. From the central values, as obtained by *the Planck collaboration*, for the cosmological redshift at matter–radiation equality $z_{eq} \equiv \frac{a_0}{a_{eq}} - 1 = 3365$, and from the present value of the ratio of the matter energy density to the critical one $\Omega_{m,0} \equiv \frac{\rho_{m,0}}{3H_0^2 M_{pl}^2} = 0.308$, one can deduce that the present value of the matter energy density is $\rho_{m,0} = 3H_0^2 M_{pl}^2 \Omega_{m,0} = 3.26 \times 10^{-121} M_{pl}^4$, and at the matter–radiation equality one should have $\rho_{m,eq} = \rho_{r,eq} = \rho_{m,0}(1 + z_{eq})^3 = 4.4 \times 10^{-1} \text{ eV}^4$, and thus, from the Stefan–Boltzmann law, $T_{eq} \sim 3 \times 10^{-28} M_{pl}$.

Since $H_0 \sim 6 \times 10^{-61} M_{pl}$, choosing a viable temperature, as, for example, $T_{rh} = 10^9 \text{ GeV}$, one has:

$$\varphi_{eq} = \varphi_{rh} + \sqrt{6} M_{pl} \quad \text{and} \quad \frac{\dot{\varphi}_{eq}}{H_0 M_{pl}} \cong 0. \tag{133}$$

That is,

$$\varphi_{eq} = \sqrt{\frac{2}{3}} M_{pl} \ln \left( \frac{3\sqrt{5}\lambda (2 - \sqrt{3}) M_{pl}^2}{2\pi T_{rh}^2} \right) + \sqrt{6} M_{pl}, \tag{134}$$

and inserting the values of $\lambda$ and $T_{rh}$, one finally obtains:

$$\varphi_{eq} \cong 27.29 M_{pl} \quad \text{and} \quad \frac{\dot{\varphi}_{eq}}{H_0 M_{pl}} \cong 0. \tag{135}$$

The Dynamical System

In order to obtain the dynamical system for this model, we introduce the following dimensionless variables:

$$x = \frac{\varphi}{M_{pl}}, \qquad y = \frac{\dot{\varphi}}{H_0 M_{pl}}. \tag{136}$$

Using the variable $N$ as time, $N = \ln\left(\frac{a}{a_0}\right)$, and from the CE $\ddot{\varphi} + 3H\dot{\varphi} + V_\varphi = 0$, one can build the following non-autonomous dynamical system:

$$\begin{cases} x' = \frac{y}{\bar{H}}, \\ y' = -3y - \frac{\bar{V}_x}{\bar{H}}, \end{cases} \tag{137}$$

where the prime means are derivative, with respect to $N$, $\bar{H} = \frac{H}{H_0}$, and $\bar{V} = \frac{V}{H_0^2 M_{pl}^2}$.

Moreover, the FE now reads:

$$\bar{H}(N) = \frac{1}{\sqrt{3}} \sqrt{\frac{y^2}{2} + \bar{V}(x) + \bar{\rho}_{rad}(N) + \bar{\rho}_{matt}(N)}, \tag{138}$$

where we have introduced the following dimensionless energy densities, $\bar{\rho}_r = \frac{\rho_{rad}}{H_0^2 M_{pl}^2}$ and $\bar{\rho}_m = \frac{\rho_{matt}}{H_0^2 M_{pl}^2}$, with:

$$\rho_{matt}(N) = \rho_{matt,eq}e^{3(N_{eq}-N)} \quad \text{and} \quad \rho_{rad}(N) = \rho_{rad,eq}e^{4(N_{eq}-N)}, \tag{139}$$

being the corresponding matter and radiation energy densities.

Integrating the dynamical system numerically, with initial conditions $x_{eq} = 27$ and $y_{eq} = 0$ obtained in (135) and imposing the condition $\bar{H}(0) = 1$, which only holds for $M \cong 10^5$ GeV (the value we have previouly obtained), we acquire the result obtained in Figure 2, where $\Omega_B = \frac{H_0^2 \bar{\rho}_B}{3H^2}$, being $B = r, m, \varphi$, is the ratio of the energy density to the critical one. We see that, at the present time $N = 0$, dark energy dominates, and that for this model, it will continue dominating forever.

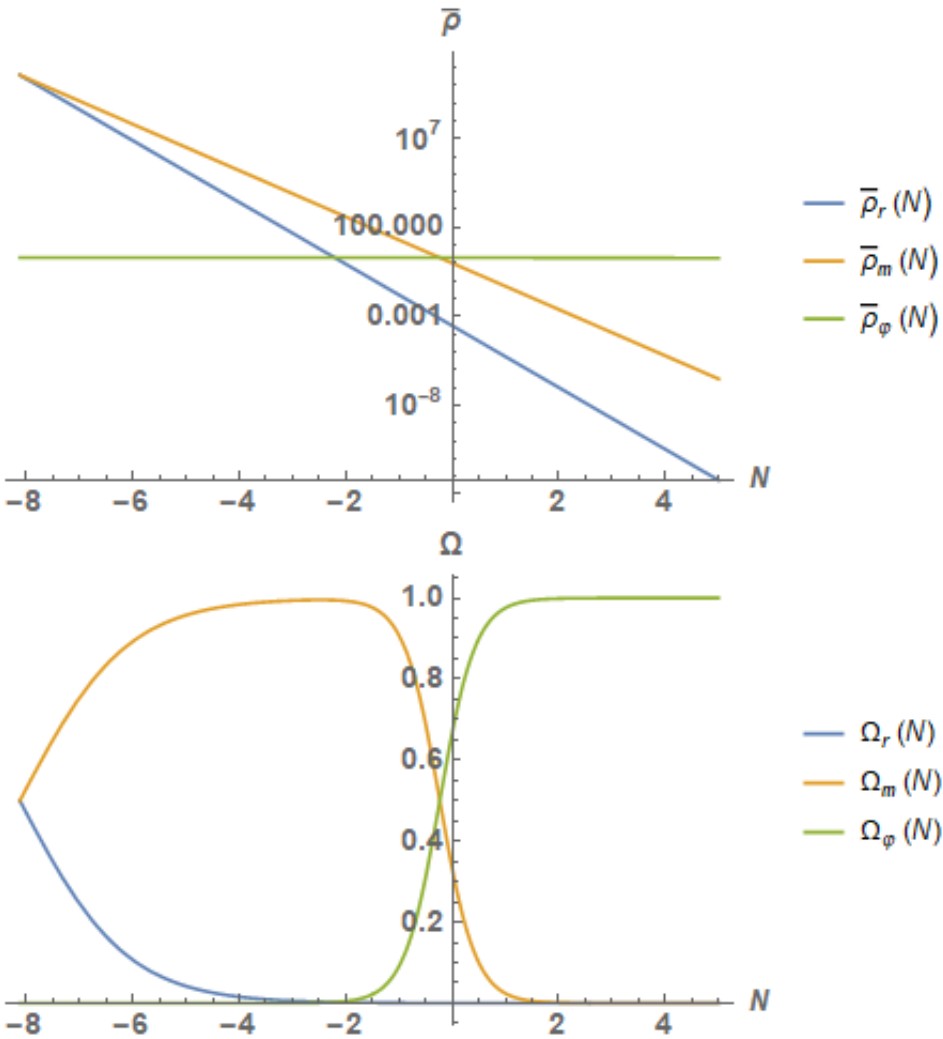

**Figure 2.** Evolution of $\{\bar{\rho}_B(N)\}_{B=r,m,\varphi}$ and $\{\Omega_B(N)\}_{B=r,m,\varphi}$.

Finally, from Figure 3, one can see the evolution of the effective EoS parameter $w_{eff} = -1 - \frac{2\dot{H}}{3H^2}$. Integration starts at the beginning of the matter–radiation equality. We see that, at present time, $w_{eff} < -1/3$, so our universe accelerates. In fact, at late times, $w_{eff}$ evolves towards $-1$, which means that the universe will always be in a state of accelerated expansion in the future.

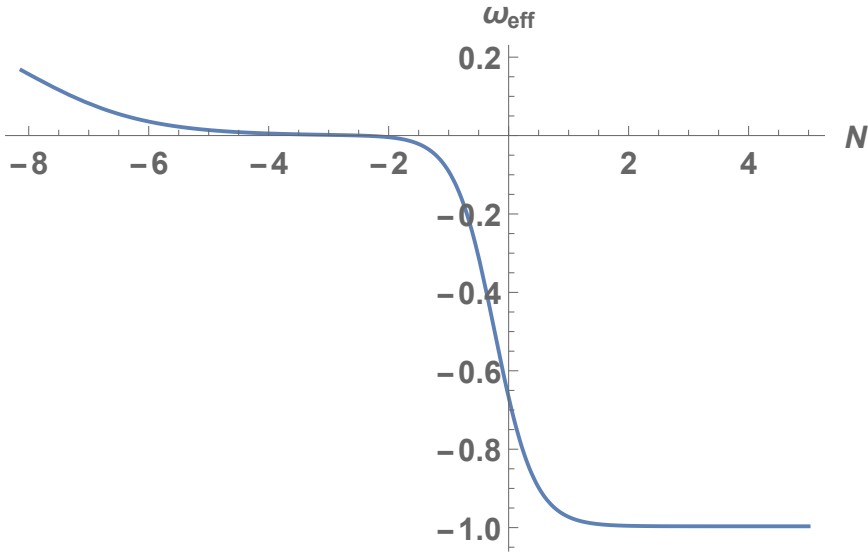

**Figure 3.** Evolution of $w_{eff}$.

## 5. The Reheating Mechanism

To understand the reheating mechanism via particle production, we shall need, first of all, some basic notions of quantum mechanics. To "warm up", the best example to study is the quantum harmonic oscillator.

### 5.1. The Harmonic Oscillator

We consider the equation of a one-dimensional harmonic oscillator:

$$\ddot{x} + \omega^2 x = 0, \tag{140}$$

whose Hamiltonian is given by:

$$\mathcal{H} = \frac{1}{2}(p^2 + \omega^2 x^2), \tag{141}$$

where $p = \dot{x}$ denotes the corresponding momentum.

The dynamical Equation (140) can be written as a Hamiltonian system:

$$\begin{cases} \dot{x} & = & \partial_p \mathcal{H} & = & \{x, \mathcal{H}\} \\ \dot{p} & = & -\partial_x \mathcal{H} & = & \{p, \mathcal{H}\}, \end{cases} \tag{142}$$

where we have introduced the Poisson bracket $\{A, B\} \equiv \partial_x A \partial_p B - \partial_p A \partial_x B$.

Next, we consider its quantum version. For this, following the correspondence principle, we need to replace the dynamical variables by the operators $x \to \hat{x}$ and $p \to \hat{p}$, and the Poisson bracket has to be replaced by a commutator, namely:

$$\{A, B\} \to -i[\hat{A}, \hat{B}] \equiv -i(\hat{A}\hat{B} - \hat{B}\hat{A}). \tag{143}$$

Thus, taking into account now that the canonically conjugate variables satisfy $\{x, p\} = 1$, one obtains its quantum analogue $[\hat{x}, \hat{p}] = i$, and using the Heisenberg picture, where the operators are time-dependent, the quantum version of the Hamilton equations read:

$$\begin{cases} \dot{\hat{x}} & = & -i[\hat{x}, \hat{\mathcal{H}}] \\ \dot{\hat{p}} & = & -i[\hat{p}, \hat{\mathcal{H}}]. \end{cases} \tag{144}$$

On the other hand, in the Schrödinger picture, which is actually equivalent to the Heisenberg one, the operators do not depend on time; thus, time dependence falls off the wave function. In fact, taking the most usual representation:

$$\hat{x} = x, \quad \hat{p} = -i\partial_x, \quad \text{and} \quad \hat{E} = i\partial_t, \tag{145}$$

the quantum version of the classical equation $E = \mathcal{H}$ (the energy being equal to the Hamiltonian) reads:

$$i\partial_t \Phi(t,x) = \frac{1}{2}\left(-\partial_{xx}\Phi(t,x) + \omega^2 x^2 \Phi(t,x)\right), \tag{146}$$

which is the well-known *Schrödinger equation* for the harmonic oscillator.

The eigenvalues of the quantum Hamiltonian are not difficult to obtain. In fact, the ground state is reached by looking for an eigenfuction of the form $e^{-ax^2}$. Inserting it in the equation $\lambda\Phi = \hat{H}\Phi$, one obtains:

$$\Phi_0(x) = \frac{1}{(\omega^2\pi)^{1/4}} e^{-\frac{\omega}{2}x^2} \text{ with } \lambda_0 = \frac{\omega}{2}, \tag{147}$$

where we have normalized the ground state.

To obtain the other eigenvalues, we diagonalize the Hamiltonian by introducing the so-called creation and annihilation operators:

$$\hat{a}^\dagger = \frac{1}{\sqrt{2\omega}}(\omega\hat{x} - i\hat{p}) \quad \text{and} \quad \hat{a} = \frac{1}{\sqrt{2\omega}}(\omega\hat{x} + i\hat{p}). \tag{148}$$

Taking into account the commutation relation between canonically conjugate variables, it is not difficult to show that the creation and annihilation operators satisfy the relation $[\hat{a}, \hat{a}^\dagger] = 1$. Then, the expression of the Hamiltonian, as a function of both operators, is:

$$\hat{\mathcal{H}} = \frac{\omega}{2}(\hat{a}\hat{a}^\dagger + \hat{a}^\dagger\hat{a}) = \omega\left(\hat{a}^\dagger\hat{a} + \frac{1}{2}\right), \tag{149}$$

where we have used the commutation relations.

Finally, with the creation operator $\hat{a}^\dagger$, we build the whole set of eigenstates:

$$\Phi_n = \frac{1}{\sqrt{n!}}(\hat{a}^\dagger)^n \Phi_0, \tag{150}$$

whose corresponding eigenvalues are $\lambda_n = \omega\left(n + \frac{1}{2}\right)$.

Now, coming back to the Heisenberg picture, the dynamical equations for the creation and annihilation operators read:

$$\dot{\hat{a}} = -i[\hat{a}, \hat{\mathcal{H}}] = -i\hat{a} \quad \text{and} \quad \dot{\hat{a}}^\dagger = -i[\hat{a}^\dagger, \hat{\mathcal{H}}] = i\hat{a}^\dagger, \tag{151}$$

and their solution is:

$$\hat{a}(t) = e^{-i\omega(t-t_i)}\hat{a}(t_i) \quad \text{and} \quad \hat{a}^\dagger(t) = e^{i\omega(t-t_i)}\hat{a}^\dagger(t_i). \tag{152}$$

Thus, one obtains:

$$\hat{x}(t) = \chi(t)\hat{a}(t_i) + \chi^*(t)\hat{a}^\dagger(t_i) \quad \text{and} \quad \hat{p}(t) = \dot{\chi}(t)\hat{a}(t_i) + \dot{\chi}^*(t)\hat{a}^\dagger(t_i), \tag{153}$$

where $\chi(t) = \frac{1}{\sqrt{2\omega}}e^{-i\omega(t-t_i)}$ is the positive frequency mode.

Finally, from the commutation relation $[\hat{x}(t), \hat{p}(t)] = 1$, one can check that the Wronskian of $\chi(t)$ and its conjugate read:

$$\mathcal{W}[\chi, \chi^*] \equiv \chi \dot{\chi}^* - \dot{\chi} \chi^* = i, \tag{154}$$

which may seem obvious for the mode $\chi(t) = \frac{1}{\sqrt{2\omega}} e^{-i\omega(t-t_i)}$, but when we deal with a time-dependent frequency, this relation will be essential.

A final remark is in order: the ground state always satisfies $\hat{a}(t)\Phi_0 = 0$, because the creation and annihilation operators evolve as (152), i.e., positive and negative frequencies do not mix. However, as we will see in the next section, for a time-dependent frequency, mixing does happen, and it is a basic ingredient that induces particle production.

*5.2. The Harmonic Oscillator with a Time Dependent Frequency*

As in the constant case, the quantum Hamiltonian is given by:

$$\hat{\mathcal{H}} = \frac{1}{2}(\hat{p}^2 + \omega^2(t)\hat{x}^2), \tag{155}$$

and the quantum equation for $\hat{x}$ in the Heisenberg picture is, again:

$$\ddot{\hat{x}} + \omega^2(t)\hat{x} = 0. \tag{156}$$

Next, we write:

$$\hat{x}(t) = \chi(t)\hat{a}(t_i) + \chi^*(t)\hat{a}^\dagger(t_i) \quad \text{and} \quad \hat{p}(t) = \dot{\chi}(t)\hat{a}(t_i) + \dot{\chi}^*(t)\hat{a}^\dagger(t_i), \tag{157}$$

where the positive frequency modes satisfy the Klein–Gordon equation $\ddot{\chi} + \omega^2(t)\chi = 0$, and also the Wronskian condition $\chi\dot{\chi}^* - \dot{\chi}\chi^* = i$. Unfortunately, these modes do not have the simple expression $\chi(t) = \frac{1}{\sqrt{2\omega}} e^{-i\omega(t-t_i)}$, as in the case of constant frequency.

In any case, we will assume that, at early times, the adiabatic condition $\frac{\dot{\omega}}{\omega^2} \ll 1$ is fulfilled, which, as we will see, does always happen during inflation. In this epoch, one can consider a positive frequency mode of the form:

$$\chi(t) = \frac{1}{\sqrt{2\omega(t)}} e^{-i \int_{t_i}^t \omega(s)ds}, \tag{158}$$

and which satisfies:

$$\dot{\chi} = \left(-i\omega - \frac{\dot{\omega}}{2\omega}\right)\chi \cong -i\omega\chi, \quad \text{and} \quad \ddot{\chi} = \left(-\frac{i\dot{\omega}}{2} - \omega^2\right)\chi \cong -\omega^2\chi. \tag{159}$$

Thus, during the adiabatic regime, the mode $\chi$ is very close to the solution that satisfies the initial conditions:

$$\chi(t_i) = \frac{1}{\sqrt{2\omega(t_i)}} \quad \text{and} \quad \dot{\chi}(t_i) = -i\sqrt{\frac{\omega(t_i)}{2}}. \tag{160}$$

Next, working in the Heisenberg picture, we consider the groundstate at time $t_i$, i.e., $\Phi_0(t_i)$, which satisfies $\hat{a}(t_i)\Phi_0(t_i) = 0$—e.g., the positive frequency mode satisfying the initial conditions (160)–and we calculate the average, at any time, of the energy with respect the ground state:

$$\Phi_0^*(t_i)\hat{\mathcal{H}}(t)\Phi_0(t_i) = \frac{1}{2}\left(|\dot{\chi}|^2 + \omega^2(t)|\chi|^2\right). \tag{161}$$

At time $t_i$, it coincides with its minimum $\omega(t_i)/2$.

Finally, we use *the diagonalization method* to look for modes of the form:

$$\chi(t) = \alpha(t)\frac{1}{\sqrt{2\omega(t)}}e^{-i\int_{t_i}^t \omega(s)ds} + \beta(t)\frac{1}{\sqrt{2\omega(t)}}e^{i\int_{t_i}^t \omega(s)ds}. \tag{162}$$

Here, $\alpha(t)$ and $\beta(t)$ are the so-called time-dependent Bogoliubov coefficients. Imposing that the mode satisfies the condition:

$$\dot{\chi}(t) = -i\omega(t)\left(\alpha(t)\frac{e^{-i\int_{t_i}^t \omega(s)ds}}{\sqrt{2\omega(t)}} - \beta(t)\frac{e^{i\int_{t_i}^t \omega(s)ds}}{\sqrt{2\omega(t)}}\right), \tag{163}$$

due to the Wronskian property $\chi\dot{\chi}^* - \dot{\chi}\chi^* = i$, they have to satisfy $|\alpha(t)|^2 - |\beta(t)|^2 = 1$. In addition, in order that the mode is a solution of the Klein–Gordon equation, these coefficients need satisfy the system of equations:

$$\begin{cases} \dot{\alpha}(t) &= \frac{\dot{\omega}(t)}{2\omega(t)}e^{2i\int_{t_i}^t \omega(s)ds}\beta(t) \\ \dot{\beta}(t) &= \frac{\dot{\omega}(t)}{2\omega(t)}e^{-2i\int_{t_i}^t \omega(s)ds}\alpha(t). \end{cases} \tag{164}$$

Inserting this expression into (161), one obtains the following "vacuum average":

$$\Phi_0^*(t_i)\hat{\mathcal{H}}(t)\Phi_0(t_i) = \frac{1}{2}\left(|\dot{\chi}|^2 + \omega^2(t)|\chi|^2\right) = \omega(t)|\beta(t)|^2 + \frac{\omega(t)}{2}. \tag{165}$$

This is an example of the well-known diagonalization method, which is used in cosmology to calculate the average energy density of the produced particles.

Generally, one subtracts the term $\omega(t)/2$, which corresponds to the minimum energy, thus obtaining:

$$\langle\hat{\mathcal{H}}(t)\rangle \equiv \Phi_0^*(t_i)\hat{\mathcal{H}}(t)\Phi_0(t_i) - \frac{\omega(t)}{2} = \omega(t)|\beta(t)|^2. \tag{166}$$

*5.3. Gravitational Particle Production of a Massive Quantum Field Conformally Coupled to Gravity*

Here, we consider a massive scalar field $\phi$ conformally coupled to gravity. Its Lagrangian density is given by [114]:

$$\mathcal{L} = \frac{1}{2}(\partial_\mu\phi\partial^\mu\phi - m_\chi^2\phi^2 - \frac{R}{6}\phi^2), \tag{167}$$

where $m_\chi$ denotes the mass and whose corresponding Klein–Gordon equation, obtained from the Euler–Lagrange equation, reads:

$$\left(-\nabla^\mu\nabla_\mu + m_\chi^2 + \frac{R}{6}\right)\phi = 0, \tag{168}$$

and, after the change of variables $\phi = \chi/a$ ($a$ denotes, once again, the scalar factor), it acquires the more usual form:

$$\chi'' - \Delta\chi + m_\chi^2 a^2\chi = 0, \tag{169}$$

where the tilde denotes the derivative with respect to conformal time, $d\tau = \frac{dt}{a(t)}$.

We now work in Fourier space, where the quantum version reads:

$$\hat{\chi}(\tau, \mathbf{x}) = \frac{1}{(2\pi)^{3/2}}\int\left(\hat{a}_{\mathbf{k}}(t_i)e^{i\mathbf{k}.\mathbf{x}}\chi_k(\tau) + \hat{a}_{\mathbf{k}}^\dagger(t_i)e^{-i\mathbf{k}.\mathbf{x}}\chi_k^*(\tau)\right)d^3\mathbf{k}, \tag{170}$$

and the modes $\chi_k$ satisfy the equation of a harmonic oscillator:

$$\chi_k'' + \omega_k^2(\tau)\chi_k = 0, \tag{171}$$

with the time-dependent frequency given by $\omega_k(\tau) = \sqrt{k^2 + a^2(\tau)m_\chi^2}$.

The corresponding average energy density is [115]:

$$\langle \rho(\tau) \rangle = \frac{1}{4\pi^2 a^2(\tau)} \int k^2 (|\chi_k'|^2 + \omega_k^2(\tau)|\chi_k|^2 - \omega_k(\tau))dk, \tag{172}$$

where, as for a single oscillator, we have subtracted the minimum vacuum energy density, i.e., the so-called zero-point oscillations of the quantum vacuum: $\frac{1}{(2\pi)^3 a^4(\tau)} \int d^3k \frac{1}{2}\omega_k(\tau)$.

At this point, it is useful once more to apply the diagonalization method, to acquire the simple form for the energy density:

$$\langle \rho(\tau) \rangle = \frac{1}{2\pi^2 a^4(\tau)} \int_0^\infty k^2 \omega_k(\tau)|\beta_k(\tau)|^2 dk, \tag{173}$$

where the time-dependent Bogoliubov coefficients, $\alpha_k$ and $\beta_k$, satisfy the system (164) for each value of $k$.

It is important to notice that $|\beta_k(\tau)|^2$ encodes the vacuum polarization effects (the creation and annihilation of pairs of opposite charge) and also the production of particles, which only happens, however, when the adiabatic evolution breaks down. Actually, in quintessential inflation, the adiabatic regime is broken during the abrupt phase transition between the end of inflation and the beginning of the kination stage and, only at the very beginning of kination is the adiabatic regime recovered. So, the polarization effects disappear and the $\beta$-Bogoliubov coefficients stabilize at a constant value, which only encodes the production of real particles.

Now, the way to calculate (numerically) $|\beta_k(\tau)|^2$ goes as follows: first of all, we need to integrate numerically the conservation equation for the inflaton field, namely,

$$\ddot{\varphi} + 3H\dot{\varphi} + V_\varphi = 0, \tag{174}$$

where $H = \frac{1}{\sqrt{3}M_{pl}}\sqrt{\frac{\dot{\varphi}^2}{2} + V(\varphi)}$, with initial conditions at some moment during the slow-roll regime. Recall that, at that instant, the system is in the slow-roll phase and, since this regime is an attractor, one only needs to take initial conditions in the basin of attraction of the slow-roll solution. Thus, we take initial conditions at horizon crossing, i.e., when the pivot scales leaves the Hubble radius $\varphi_*$ and $\dot{\varphi}_* = -\frac{V_\varphi(\varphi_*)}{3H_*}$.

Once we have obtained the evolution of the background, and, in particular, the evolution of the Hubble rate, we compute the evolution of the scale factor. This is given by:

$$a(t) = e^{\int_{t_*}^t H(s)ds}, \tag{175}$$

where we have chosen, at the horizon crossing, $a_* = 1$. Once we have the evolution of this scale factor, we just need to numerically integrate the system (164).

A final remark is in order. When particles are produced, the Friedmann equation is modified as follows:

$$H^2 = \frac{1}{3M_{pl}^2}(\rho_\varphi + \langle \rho \rangle), \tag{176}$$

where $\rho_\varphi$ denotes the energy density of the inflaton field and $\langle \rho \rangle$ denotes the one of the produced particles.

### 5.4. Reheating in the Case of α-Attractors

Dealing with α-attractors (see the potential (31)), one can show that superheavy particles, with masses around $m_\chi \sim 10^{15}$ GeV, are gravitationally produced during the abrupt phase transition from the end of inflation to the beginning of kination. Note that such heavy masses are needed so that the polarization effects do not affect the dynamics of the inflaton field during inflation, because it is usually assumed that inflation starts at GUT scales with $H_{GUT} \sim 10^{14}$ GeV, and it is well known that polarization effects can be neglected when $H \ll m_\chi$.

Therefore, the produced χ-particles have energy densities of the order (see for instance [116]):

$$\langle \rho(\tau) \rangle \sim \frac{m_\chi^4}{6\pi^2} e^{-\kappa \alpha m_\chi / H_*} \left( \frac{a_{kin}}{a(\tau)} \right)^3, \tag{177}$$

where κ is a dimensionless number of order 1 (more precisely, for $\alpha = 10^{-3}$, numerically, one obtains $\kappa \cong 12.13$; for $\alpha = 10^{-2}$, the calculation yields $\kappa \cong 3.45$; for $\alpha = 10^{-1}$, one obtains $\kappa \cong 1.1$; and, finally, for $\alpha = 1$, one gets $\kappa \cong 0.3$), and $H_* \sim 10^{13}$ GeV is the escale of inflation, i.e., the value of the Hubble parameter at horizon crossing.

What is actually important is that these superheavy particles all decay into lighter ones to form a relativistic plasma which will eventually dominate the evolution of the universe and, thus, conveniently match the hot Big Bang model. Then, two different situations may arise:

1. The decay takes place before the end of the kination phase (recall that kination ends when $\rho_\varphi \sim \langle \rho \rangle$);
2. The decay occurs after the end of kination.

Let Γ be the decay rate; then, the decay is finished when $\Gamma \sim H$, because $H \sim 1/t$, the number of χ-particles, decays as $e^{-\Gamma t}$, and the decay is practically finished when $e^{-\Gamma t} \cong 1/2$.

#### 5.4.1. Decay Happens before the End of Kination

In this case, we have the following two constraints:

1. The decay occurs after the beginning of kination, i.e., $\Gamma \leq H_{kin} \sim 6 \times 10^{-7} M_{pl}$. This value has been obtained numerically in the case of α-attractors, and it agrees with the fact that kination starts immediately after the end of inflation, which, in the majority of models, ends at the scale $H_{END} \sim 10^{-7}$ GeV;
2. The decay precedes the end of kination, i.e., $\langle \rho_{dec} \rangle \leq \rho_{\varphi,dec}$. Taking into account that the energy density of the background, i.e., the one of the inflaton field, and the one of the relativistic plasma, when the decay is finished, that is, when $\Gamma \sim H_{dec} = H_{kin} \left( \frac{a_{kin}}{a_{dec}} \right)^3$, will be:

$$\rho_{\varphi,dec} = 3\Gamma^2 M_{pl}^2 \quad \text{and} \quad \langle \rho_{dec} \rangle = \langle \rho_{kin} \rangle \left( \frac{a_{kin}}{a_{dec}} \right)^3 \cong \frac{m_\chi^4}{6\pi^2} e^{-\kappa \alpha m_\chi / H} \frac{\Gamma}{H_{kin}}, \tag{178}$$

one can see that these two constraints bound the decay rate, as follows:

$$\frac{1}{18\pi^2} e^{-\kappa \alpha m_\chi / H_{inf}} \frac{m_\chi^4}{H_{kin} M_{pl}^2} \leq \Gamma \leq H_{kin}. \tag{179}$$

Using, once again, *Stefan–Boltzmann's* law $\rho_{rh} = \frac{\pi^2 g_{rh}}{30} T_{rh}^4$, with $g_{rh} \cong 106.75$ (the degrees of freedom for the standard model), the reheating temperature is given by:

$$
\begin{aligned}
T_{reh} &= \left( \frac{30}{\pi^2 g_{reh}} \right)^{1/4} \langle \rho_{reh} \rangle^{\frac{1}{4}} = \left( \frac{30}{\pi^2 g_{reh}} \right)^{1/4} \langle \rho_{dec} \rangle^{\frac{1}{4}} \sqrt{\frac{\langle \rho_{dec} \rangle}{\rho_{\varphi,dec}}} \\
&\cong \sqrt{3\pi} \left( \frac{30}{\pi^2 g_{reh}} \right)^{1/4} \times e^{-\frac{3}{4}\kappa\alpha m_\chi / H_{inf}} \left( \frac{6 H_{kin}}{\Gamma} \right)^{1/4} \frac{m_\chi^3}{M_{pl}^2 H_{kin}} M_{pl},
\end{aligned}
\tag{180}
$$

where we have it that reheating happens when $\langle \rho_{reh} \rangle \sim \rho_{\varphi,reh}$, and recalling that the evolution of both energy densities is given by:

$$
\langle \rho_{reh} \rangle = \langle \rho_{dec} \rangle \left( \frac{a_{dec}}{a_{rh}} \right)^4 \quad \text{and} \quad \rho_{\varphi,reh} = \rho_{\varphi,dec} \left( \frac{a_{dec}}{a_{rh}} \right)^6,
\tag{181}
$$

which means that $\left( \frac{a_{dec}}{a_{rh}} \right)^2 = \frac{\langle \rho_{dec} \rangle}{\rho_{\varphi,dec}}$, and thus:

$$
\langle \rho_{reh} \rangle = \frac{\langle \rho_{dec} \rangle^3}{\rho_{\varphi,dec}^2}.
\tag{182}
$$

Finally, taking now into account the bound (179), we obtain that, for $\alpha \sim 10^{-2}$, the reheating temperature ranges between $10^7$ GeV and $10^9$ GeV. It is interesting to compare this enormously high reheating temperature with the solar surface temperature, which is around 6000 K $\sim 8 \times 10^{-11}$ GeV (although it is a couple of orders of magnitude higher in its interior).

5.4.2. Decay after the End of Kination

In this situation, we have the constraint $\Gamma \leq H_{end}$, where $H_{end}$ denotes the end of kination. Taking this into account, one obtains:

$$
H_{end}^2 = \frac{2\rho_{\varphi,end}}{3M_{pl}^2}
\tag{183}
$$

and

$$
\rho_{\varphi,end} = \rho_{\varphi,kin} \left( \frac{a_{kin}}{a_{end}} \right)^6 = \frac{\langle \rho_{kin} \rangle^2}{\rho_{\varphi,kin}},
\tag{184}
$$

where we have that kination ends when $\langle \rho_{end} \rangle \sim \rho_{\varphi,end}$, what means that $(a_{kin}/a_{end})^3 = \frac{\langle \rho_{kin} \rangle}{\rho_{\varphi,kin}}$, because the energy density of the inflaton decays as $a^{-6}$, but the one of matter decays as $a^{-3}$. So, the condition $\Gamma \leq H_{end}$ leads to the bound:

$$
\Gamma \leq \frac{\sqrt{2}}{18\pi^2} e^{-\kappa\alpha m_\chi / H_{inf}} \frac{m_\chi^4}{H_{kin} M_{pl}^2}.
\tag{185}
$$

Assuming, as usual, that thermalization is nearly instantaneous, reheating occurs when the decay is finished, i.e., when $\Gamma \sim H$, and thus, $\langle \rho_{dec} \rangle \sim 3\Gamma^2 M_{pl}^2$, which leads to the following reheating temperature:

$$
T_{reh} = \left( \frac{30}{\pi^2 g_{reh}} \right)^{1/4} \langle \rho_{dec} \rangle^{1/4} = \left( \frac{90}{\pi^2 g_{reh}} \right)^{1/4} \sqrt{\Gamma M_{pl}};
\tag{186}
$$

using the previous bound, one obtains that, when the decay of the superheavy particles occurs after the end of the kination phase, the reheating temperature belongs in the range $1 \text{ MeV} \leq T_{rh} \leq 10^8 \text{ GeV}$.

## 6. Historical Notes

1.  On 20 November 1915, Hilbert gave a talk at the Royal Society of Sciences in Göttingen, which was later published in the Transactions of the Society, in March 1916. There, Hilbert presented the covariant equations for GR. On the other hand, Einstein's presentation, for the first time, of his equations for GR took place in the Prussian Academy of Sciences in Berlin on 25 November 1915, five days after Hilbert's.

    So, at first glance, it would seem that it was Hilbert who first obtained these equations. This was indeed the viewpoint of some scientists contemporary to Einstein and Hilbert, among them Felix Klein, Wolfgang Pauli, and Herman Weyl. In fact, for years there was an ongoing controversy about whether it was Albert Einstein or David Hilbert who had first obtained the GR equations.

    However, a new document appeared later in Hilbert's archive at the University of Göttingen: the printing proofs of the first version of Hilbert's paper, published in March of 1916. These proofs were sent to Hilbert two weeks after his talk (on 6 December 1915), and there, one can check that Hilbert did not present the equations of GR in his talk of November 1915. Quite on the contrary, in these proofs, Hilbert refers explicitly to Einstein's talk of 25 November, published on 2 December 1915. What really happened is that Hilbert included the GR equations in his publication, which he obtained in an alternative way, but only after reading Einstein's paper, having made sure to check that the results coincided.

    More to the point, Einstein wrote to Hilbert: *"I had no difficulty finding the general covariance equations of GR. This is easy with the help of the Riemann tensor. What is really difficult is to recognize that these equations constitute a generalization, and even more, a simple and natural generalization of Newton's laws"*.

    What is also clear is that Hilbert discussed GR in a superficial way only, concentrating on the mathematical structure of the equations and on their Lagrangian formulation, but probably without understanding in depth their physical meaning, quite the opposite of Einstein's approach to this issue.

2.  Astronomers who made most important contributions to understand the expansion of our universe were Vesto Slipher, Henrietta Leavitt, and Edwin Hubble. However, the first person who clearly realized that the universe is expanding was a Belgian priest, mathematician, and physicist named Georges Lemaître, who published his results in 1927. We should note that, in those years, everybody believed that the universe was static, and for very strong physical reasons. Indeed, as any ordinary physical system that had more than enough time to evolve (an infinite amount of time, in theory, since the universe was considered to have always existed), it should have necessarily reached the stationary state. It could not be otherwise. However, Lemaître proved reality to be quite the contrary. On the basis of Einstein's GR and by matching the theory with the astronomical observations of Slipher and Hubble, in a masterful way, he proved that the universe was expanding; later, he observed that it was not eternal, that it had an origin.

    For many decades (even now, it is still so declared in most places) people believed that the astronomer Edwin Hubble was the person who first discovered the expansion of the universe. Only recently, without denying at all Hubble's important contributions, have historians put things in the right place. A very detailed account of this thrilling story can be found in a book recently published by one of the authors [1].

3.  In his calculations, using his table of distances (obtained in part with the help of Leavitt's law), and Slipher's table of velocities (obtained as optical Doppler shifts), Hubble obtained a rather large value for the expansion rate, of $H_0 \cong 500 \frac{\text{km/s}}{\text{Mpc}}$, which is off the presently accepted value by almost one order of magnitude. The reason is that

it is extremely difficult to measure cosmological distances. On the contrary, obtaining velocities by means of the Doppler shifts is somehow easier. However, here there is also the problem of appropriately disentangling the recession redshift from other contributions to the observed Doppler shift, coming from the gravitational influence of other massive celestial objects. Even today, there is still a sharp controversy about the right value of $H_0$ (see, e.g., [57] and references therein). Results from astronomical observations by different groups, each one reporting uncertainties of just 1 or 2%, differ by some 5 to 10%, an unpleasant situation that is termed the "Hubble parameter tension" [58,59].

4.  In 1922, Alexander Friedmann was the first to discover full families of solutions of the EEs, which he rightly interpreted as corresponding to expanding and to contracting universes. In 1922 and 1924, Friedman published two seminal papers in the prestigious German journal *Zeitschrift für Physik* [7], of which Einstein was an editor. In those papers, Friedmann showed that there were solutions to Einstein's equations where the universe evolved in an expanding or contracting way. Recall (see above) that, in this epoch, everybody believed that our universe was static. However, Friedmann explicitly declared, in 1924, that, based on some of his solutions, our universe very well might be expanding.

    Einstein was the "referee" (this figure did not actually have the same meaning and function at that time) of Friedmann's first paper and, after having studied it, he wrote a letter to the journal saying that Friedmann's calculations should not be published since they contained an error. When Friedmann (indirectly) learned of this opinion, he sent all the details of his calculation to Einstein, asking him to check them for himself. After some discussions (described in detail in, e.g., [1–4]), Einstein finally recognized that Friedmann made no mistake, and hurried to publish another letter recognizing his own error and saying that Friedmann's paper should be published. Unfortunately, in those days, it took a long time to see an article published after it was finished, and Friedmann died in 1925 before the publication of Einsteins' retraction.

5.  During a break at the very famous Solvay meeting of 1927 in Brussels, Einstein said, literally, to Lemaître (as was reported later by the last): "Vos calculs sont corrects, mais votre physique est abominable". He was referring to Lemaître's paper of the same year (1927) [9], which he had handed to Einstein during a previous conversation. In the paper (in French), Lemaître had obtained, for the first time ever, the Hubble law, and moreover, he had interpreted it in the right way, as being a proof of the expansion of the universe, of the very fabric of the cosmos (an interpretation that Hubble never admitted, in his whole life). Einstein himself did not accept the expansion of the cosmos until 1932, when he was finally convinced by Richard Tolmann and Willem de Sitter [1–4].

6.  In 1956, George Gamow wrote, in *Scientific American*, that Einstein had told him, long ago, that the idea of the cosmic repulsion associated with the cosmological constant had been "the greatest blunder of his life" ("Die grösste Eselei meines Lebens", in German). For years to come, this was the only testimony of such a claim, leading many to question it because of Gamow's well-known imaginative character. Recently, however, it has been discovered by historians of physics ([117] and references therein) that Einstein made a similar statement on at least two more occasions. Indeed, John Wheeler wrote in his book, *Exploring Black Holes: Introduction to General Relativity*, that he had personally been present when Einstein said the above words to Gamow, outside the hall of the Institute for Advanced Studies in Princeton. Moreover, Ralph Alpher also testified once that he had heard Einstein make such a claim.

    In addition, it is a proven fact that Einstein never wanted to use the CC again, not even when someone suggested that it might be interesting to put it back, so as to better adjust the age of the universe to the results of observations of the oldest galaxies, which seemed, at one point (erroneously), to clearly exceed the age of the universe. As Einstein explained in a footnote in the appendix to the second edition of his book

*The Meaning of Relativity* [118]: "If Hubble's expansion had been discovered at the time of the creation of the general theory of relativity, the cosmological constant would never had been added. It now seems much less justified to add a term like this in the field equations, since its introduction loses the only justification it originally had". Finally, an important consideration, which very few mention, is the following [1]: in taking this position, Einstein was even more radical than Friedmann and Lemaître (the defendants of the expanding universe), since those always included the CC term in their models for the universe; even if such a term *was not necessary* at all in their equations, contrary to the case of Einstein's static model, where it was crucial (for an expanding universe solution, such an additional term plays a secondary role). In any case, if there was no reason for its presence, it should not be put there, under any circumstance; this is what Einstein said (please see [1] for additional details).

7. Fred Hoyle, an English nuclear physicist and astronomer who formulated the theory of stellar nucleosynthesis (and, with it, the remarkable fact that we are all stardust), was one of the authors of the *steady-state theory* of the universe, an attempt to maintain a static model for the cosmos that is able to account for Hubble's empirical law of expansion [1–4]. Hoyle did not buy Lemaître's conclusion that the universe had an origin, much less his *hypothesis of the primeval atom* that latter exploded; he, as a serious nuclear physicist, understood it as lacking any physical rigor. On BBC Radio's Third Programme broadcast on 28 March 1949, Hoyle explained to the audience that, when comparing Lemaître's model (by then improved by Gamow) with his steady-state theory (where a smooth creation of matter had to take place in order to compensate for Hubble's expansion and keep the matter density of the universe constant), in Lemaître's model a sudden creation of all the matter in the universe had to occur at the very beginning of it. For this to happen, an unbelievably huge expansion (a Big Bang) of the fabric of space was absolutely necessary. However, of course, such a phenomenon was fully impossible, and therefore, he pronounced these famous words in a very disdainful tone (see a much more detailed explanation in [1]). Hoyle anticipated the idea of cosmic inflation very clearly, albeit as an impossible thought, exactly thirty years before Alan Guth, on an inspired night, could formulate it precisely.

 However, even if Hoyle had spoken these two words, Big Bang, in a disrespectful manner—trying, on purpose, to mock Lemaître's model (which had, by then, substantially been improved by George Gamow)—from this moment on, everybody, starting with Gamow himself (a very peculiar character, as is well known) began to use this term to refer to the origin of the universe.

8. The cosmological horizon problem (aka the homogeneity problem) is a fine-tuning issue that affects classical Big Bang models of the universe. It arises due to the impossibility of explaining the homogeneity reported by astronomical surveys of very distant regions of space—which are causally disconnected in these Big Bang models—unless one invokes a mechanism that sets the same initial conditions everywhere with very high accuracy. This problem was first pointed out by Wolfgang Rindler in 1956. The most commonly accepted solution is cosmic inflation, as we have discussed here, but an explanation in terms of a variable speed of light has been proposed, too.

9. The flatness problem is another important issue that appeared in the old, classical Big Bang model of the universe. It was first mentioned by Robert Dicke in 1969, in the Jane Lectures he gave for the American Philosophical Society that year. The total normalized energy density of our present universe has been measured to be very close to 1, with very small uncertainty, which points towards a very flat universe. Any departure from the conditions leading to this value in the past would had been magnified enormously over cosmic time. This leads to the conclusion that one would need an unbelievably accurate fine-tuning in the initial conditions of the universe, with an energy density that should have been incredibly closer to the critical value at the very beginning of the universe. As we have seen, cosmic inflation

provides the solution to this problem by making the universe extremely flat, to the needed precision.

## 7. Conclusions

Our main aim in this review was to explain important issues in modern cosmology in a simple and comprehensible way. To start, from Hubble's law and by using a "homogeneous" version of the Einstein–Hilbert action, together with the first law of thermodynamics, we have easily derived the constituent equations of cosmology. Then, we dealt with their mathematical singularities, namely, the famous Big Bang singularity and some possible future singularities, such as the Big Rip and others, providing different ways to remove them and eventually obtain physically sound results.

Next, we have explained a number of shortcomings that had appeared in the old Big Bang cosmology, such as the horizon and the flatness problems, whose solutions were given in terms of the inflationary paradigm introduced by Alan Guth. We have reviewed inflation in detail and in very understandable terms. We have focused on the slow-roll regime, explicitly showing its attractor behavior and its precise relation to the slow-roll parameters. We have also calculated, step by step, the number of e-folds the universe must necessarily expand by, in order to overcome all these problems.

Later, we focused on the study of the current cosmic acceleration via the introduction of different sorts of dark energy. Specifically, we first considered the most simple model, which uses the cosmological constant. Here we have shown, as a warm up exercise, that the original approach by Einstein—namely, his static model of the universe—was unstable (in fact, we show in detail that it corresponds to a saddle point of the model, viewed as a dynamical system). Then, we have considered a quintessence field, which can also be used to unify both periods of inflation, under the form of a most popular theory, named quintessential inflation, which we have discussed in some detail.

We have also analyzed the universe's reheating mechanism, which is a very important and necessary stage after inflation ends, by analyzing, again, a basic system: the quantum harmonic oscillator with a time-dependent frequency. Within this simple example, we have introduced the standard diagonalization method, based on the calculation of the time-dependent Bogoliubov coefficients. As an application, we have derived the bounds of the reheating temperature for the model of an $\alpha$-attractor in the context of quintessential inflation.

To finish, in a closing section, we have added a few short notes, which provide updated descriptions of a number of important historical events. A much more detailed account of them is given in [1], which is a perfect complement to the present, more technical, albeit still very pedagogical, review.

**Author Contributions:** J.d.H. and E.E. have contributed to this paper in the investigations, formal analysis, resources acquiring, and final writing, review, editing and supervision. All authors have read and agreed to the published version of the manuscript.

**Funding:** J.d.H. is supported by grant MTM2017-84214-C2-1-P, funded by MCIN/AEI/10.13039/501100011033, and by "ERDF A way of making Europe". E.E. is supported by MICINN (Spain), project PID2019-104397GB-I00, of the Spanish State Research Agency program AEI/10.13039/501100011033, and by the program Unidad de Excelencia María de Maeztu CEX2020-001058-M. Both authors are also supported in part by the Catalan Government, AGAUR project 2017-SGR-247.

**Institutional Review Board Statement:** The study was conducted in accordance with the Declaration of Helsinki. Ethical approval is not applicable for this study.

**Informed Consent Statement:** Not applicable.

**Data Availability Statement:** Not applicable.

**Conflicts of Interest:** The funders had no role in the design of the study; in the collection, analyses, or interpretation of data; in the writing of the manuscript, or in the decision to publish the results.

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
