# Peer review of "Topics in Cosmology—Clearly Explained by Means of Simple Examples"

_universe, doi:10.3390/universe8030166_

Round 1

Reviewer 1 Report

Dear Sirs,

the manuscript “Some topics in Cosmology - Clearly explained by means of simple examples” by Jaume de Haro, Emilio Elizalde

reviews some aspects of cosmology. In particular, the authors review the inflationary scenario and the reheating mechanisms, and they present the scenario of so-called quintessential inflation in which the two phases of Universe acceleration can be described in a unified way. The motivation is good, the analysis and presentation has been performed in a correct way, and the paper contains results and information that although are not new, they are interesting for the community. However, before I can recommend publication the following minor issues need to be handled:

1) The authors should add some comments on the minimum values of the tensor-to-scalar ratio that can be obtained in this scenario. Can the \alpha-attractors lead to quite small r values?

2) The authors should add some comments on the possible couplings of neutrinos. Isn’t it necessary for obtaining a successful quintessential inflation, as it was shown in [77], or one can obtain a successful quintessential inflation without it?

3) In the end of page 43 it is written that Lemaître proved that the Universe has an origin. However, Lemaître did not “proved” that, nor there is such a proof as far as I know. In fact it is a common consensus in gravity and cosmology that we cannot describe things above the Planck scale since we do not know the corresponding physics. In standard language, we cannot describe the physics that happened in the era before the Universe was at Planck length, and therefore we cannot conclude and prove that there was an origin or not.

4) The authors could add a comment that unified scenarios,from inflation to late-time acceleration, can be also obtained in modified gravity, either in f(R) gravity or in f(T) gravity.

When these issues will have been handled, I will recommend the paper for publication in the Universe.

Author Response

Our answer is attached in a file

Reviewer 2 Report

The authors presented a very complete and comprehensive review of standard cosmology. The article is really well written, easy to read and understand.

However, I would suggest removing the manuscript from the article section of the Universe Journal and inserting it in the Review section.

Additionally, it would be interesting for the reader to have some more in-depth information on the state of the art from an observational and statistical point of view.

For example, when the authors talk about the tension relative to the Hubble parameter, it would be interesting for the reader to have more information regarding the nature of the tension (i.e. late-time versus early-time data).

Another example is that the hypothesis of homogeneity and isotropy is extremely well supported by surveys of galaxies (e.g. SDSS) and by the recent results of WMAP and Planck satellites.

Finally, a mention of the importance of dark matter on a cosmological level (e.g. formation of large-scale structures) I believe would give a better picture of the current state of the art of the BB cosmology.

Author Response

We have attached our answer in a pdf file

Reviewer 3 Report

The paper is a partial description of the modern cosmology. The examples and the approach are very interesting. Therefore I recommend the publication.

Author Response

Thank you for the comments